# Functional and structural diversity in deubiquitinases of the *Chlamydia*-like bacterium *Simkania negevensis*

Vanessa Boll [1], Thomas Hermanns [1], Matthias Uthoff[2,4], Ilka Erven[1], Eva-Maria Hörner [3], Vera Kozjak-Pavlovic [3], Ulrich Baumann [2] & Kay Hofmann [1]✉

Besides the regulation of many cellular pathways, ubiquitination is important for defense against invading pathogens. Some intracellular bacteria have evolved deubiquitinase (DUB) effector proteins, which interfere with the host ubiquitin system and help the pathogen to evade xenophagy and lysosomal degradation. Most intracellular bacteria encode one or two DUBs, which are often linkage-promiscuous or preferentially cleave K63-linked chains attached to bacteria or bacteria-containing vacuoles. By contrast, the respiratory pathogen *Legionella pneumophila* possesses a much larger number of DUB effectors, including a K6-specific enzyme belonging to the OTU family and an M1-specific DUB uniquely found in this bacterium. Here, we report that the opportunistic pathogen *Simkania negevensis*, which is unrelated to *Legionella* but has a similar lifestyle, encodes a similarly large number of DUBs, including M1- and K6-specific enzymes. *Simkania* DUBs are highly diverse and include DUB classes never before seen in bacteria. Interestingly, the M1- and K6-specific DUBs of *Legionella* and *Simkania* are unrelated, suggesting that their acquisition occurred independently. We characterize the DUB activity of eight *Simkania*-encoded enzymes belonging to five different DUB classes. We also provide a structural basis for the M1-specificity of a *Simkania* DUB, which most likely evolved from a eukaryotic otubain-like precursor.

The covalent attachment of ubiquitin to intracellular proteins is a versatile posttranslational modification pathway found throughout eukaryotes. In a multistep mechanism involving E1, E2, and E3 enzyme components, the C-terminus of ubiquitin is activated and transferred onto a lysine side chain of the target protein, forming an isopeptide bond. The complexity of ubiquitin-based signals is further increased by the propensity of ubiquitin to become ubiquitinated on one or more of its lysine residues, leading to the formation of ubiquitin chains. Depending on the modified lysine, ubiquitin chains of different linkage types are formed, which confer different fates to the modified protein[1].

Recent studies have revealed further layers of complexity through the use of mixed and branched chains, ubiquitination of non-lysine residues, and ubiquitination of non-protein targets[2,3]. Among the most important linkage types are K48-linked chains targeting proteins for proteasomal degradation and K63-linked chains regulating DNA repair, endocytosis, and vesicular trafficking. Linear chains, also known as M1-linked chains, are formed by ubiquitination of the ubiquitin N-terminus and are important for NF-κB activation and immune signaling.

Typical bacteria and archaea do not have a ubiquitin system, although they encode distantly related proteins that are not used for

[1]Institute for Genetics, University of Cologne, Cologne, Germany. [2]Institute of Biochemistry, University of Cologne, Cologne, Germany. [3]Chair of Microbiology, Biocenter, Julius Maximilian University, Würzburg, Würzburg, Germany. [4]Present address: Bayer AG, Research & Development, Pharmaceuticals, Biologics Research, Wuppertal, Germany. ✉e-mail: kay.hofmann@uni-koeln.de

protein modification[4]. However, most intracellular bacteria have to deal with the host ubiquitin system, which is used by many species as an antibacterial defense mechanism[5]. After cell entry by endocytosis, some intracellular bacteria live in specialized bacteria-containing vacuoles (BCVs), whereas other bacteria escape into the cytoplasm[6,7]. Both free bacteria and BCVs are subject to ubiquitination by the host cell, targeting them for clearance by xenophagy[8]. Several intracellular bacteria have evolved effector proteins which are secreted into the host cytoplasm and counteract ubiquitin-based defenses, for example, by preventing ubiquitination[9–11], removing ubiquitin chains[12,13], or interfering with ubiquitin-induced autophagy[14,15].

The most numerous of these effectors are bacterial deubiquitinases, which have been identified and characterized in many important intracellular pathogens. Examples include SseL from *Salmonella typhimurium*[16], ChlaDUB1/2 from *Chlamydia trachomatis*[17], ChlaOTU from *Chlamydia pneumoniae*[18], TssM from *Burkholderia pseudomallei*[19], and several deubiquitinases from *Legionella pneumophila*[20]. Most intracellular bacteria code for one, sometimes two, DUB effectors, which typically show little linkage specificity. By contrast, *L. pneumophila* is unique among characterized bacteria in that it encodes multiple DUBs[20,21], including the linear chain (M1)-specific RavD[22] and the K6-specific LotA[23,24]. The majority of bacterial DUBs belong to two different protease families: the ovarian-tumor related (OTU) family, known for eukaryotic linkage-specific DUBs[25], and the CE-clan, whose eukaryotic members are proteases for ubiquitin-like modifiers, but no DUBs[26]. The only known exceptions are *Burkholderia* TssM, which is a member of the eukaryotic USP (ubiquitin-specific protease) family[19], and *Legionella* RavD, which is not obviously related to any eukaryotic enzyme family[22]. Other DUB families, such as UCH, Josephin, MINDY, and ZUFSP, are not known to have non-eukaryotic members.

Recently, we performed a bioinformatical analysis of deubiquitinase sequences and structures, aiming at the identification of features that discriminate enzymes with DUB activities from other papain-fold cysteine proteases[21]. Based on these findings, we established a bioinformatics pipeline for predicting DUBs from bacterial genome sequences[27]. While applying these methods to genomes of potential intracellular pathogens, we noticed that the *Chlamydia*-like bacterium *Simkania negevensis* encodes a particularly large deubiquitinase complement, which rivals that of *Legionella* in number and complexity. *S. negevensis* belongs to the *Chlamydiales*, part of a phylum of gram-negative bacteria distant from the *Legionellales*, which belong to the γ-proteobacteria. However, *L. pneumophila* and *S. negevensis* have similar lifestyles; they proliferate in bacteria-containing vacuoles inside the host cell cytoplasm and can infect a wide range of hosts, including amoebae and human cells[28,29]. *S. negevensis* is characterized by a protracted life cycle; the development of the inclusion membrane and bacterial proliferation reaches a plateau after 3 days, whereas cell lysis and the release of the infectious bacterial form begin at day 4 post infection[30].

In this work we identified twelve *Simkania* DUB candidates and found deubiquitinating activity for eight of them. Interestingly, *Simkania* possesses DUB families not commonly observed in bacteria, including three USP-type deubiquitinases, two members of the Josephin family, and one member of the recently identified viral-tegument like deubiquitinase (VTD) family[31]. Similar to *L. pneumophila*, *S. negevensis* possesses strictly K6- and M1-specific DUBs, although they belong to different classes and appear to be independent acquisitions from their eukaryotic hosts. By solving the structure of the *Simkania* M1-specific DUB in complex with di-ubiquitin, we demonstrate the basis for its linear specificity to be distinct from that of OTULIN[32] and RavD[22], the other two linear-specific DUBs known to date.

## Results
### Discovery of DUB candidates in *Simkania negevensis*
Following the bioinformatical DUB discovery pipeline described earlier[27], we applied two major approaches. First, generalized profiles[33] derived from established families of DUBs and related proteases were searched against the *S. negevensis* proteome. In a second, more sensitive approach, Hidden–Markov–Models (HMMs) of established DUB families were searched against a database of pre-calculated HMMs generated from each *S. negevensis* ORF and its homologs from other species[34]. Significant hits in either search were subjected to further tests, for example, by assessing the presence of a DUB-typical conserved gatekeeper motif[21], and by testing if the hits are more closely related to a DUB family than to other cysteine protease families covered in the MEROPS database[35]. Using this approach, we identified three ORFs with obvious similarity to the eukaryotic USP family. In all three cases (SNE_A12110/SnUSP1, SNE_A12380/SnUSP2, and SNE_A05310/SnUSP3), the region of detectable conservation covers the entire USP domain; all canonical active site residues are conserved and the expected aromatic gatekeeper residues are found adjacent to the catalytic His residues (Supplementary Fig. 1a). HMM-to-HMM searches uncovered significant similarities between the two *S. negevensis* ORFs and Josephin (ATAXIN3-like) DUB domains. In both cases (SNE_A21920/SnJos1 and SNE_A21910/SnJos2), the conserved region spans the entire DUB domain, including all canonical active site residues. Surprisingly, SnJos1 and SnJos2 lack aromatic gatekeeper motifs (Supplementary Fig. 1b). A further *S. negevensis* DUB candidate was closely related to members of the newly described Viral tegument-like deubiquitinase (VTD) family[31]. The entire catalytic domain and all active site residues are conserved in SNE_A13000/SnVTD. Due to their inverted active site, VTD deubiquitinases do not rely on aromatic gatekeeper residues[31] (Supplementary Fig. 1c). Only a single *S. negevensis* ORF matched the OTU deubiquitinase family: SNE_A17630/SnOTU shows conservation throughout the catalytic domain, including the active site residues and gatekeeper motif (Supplementary Fig. 1d) conservation. Finally, five *S. negevensis* ORFs were shown to be related to CE-clan enzymes using HMM-to-HMM searches. One of these hits, SNE_A10940/SnCE1, appears to be a homolog of *Chlamydia* deubiquitinase ChlaDUB1/CDU1[17], and the other four (SNE_A13010/SnCE2, SNE_A19290/SnCE3, SNE_A14650/SnCE4, and SNE_A22800/SnCE5) show no particular similarity to any established CE-clan DUB (Supplementary Fig. 1e). No significant similarities to UCH, MINDY, or ZUFSP-type deubiquitinases were detected.

Of all the DUB candidates, only SNE_A05310/SnUSP3 had been identified in a proteomic analysis of purified *Simkania*-containing vacuole membranes[36]. To investigate the mRNA levels of the bioinformatically predicted DUBs during the infection cycle in human HeLa cells and *Acanthamoeba castellanii*, we performed quantitative PCR analysis (Supplementary Fig. 2). PCR products were detected for all genes tested; a comparison of their relative intensities after normalization by bacterial 5 S rRNA revealed a wide range of expression levels. Measurements in HeLa cells (Supplementary Fig. 2a) showed that all DUBs had much weaker expression than the bacterially-encoded protein GroEL, with the strongest signals observed for SnCE1, SnCE2, SnOTU, and SnCE3. Several candidates, including SnUSP1, SnUSP2, SnJOS1, SnJOS2, SnCE2, and SnCE4, showed a moderate upregulation in the later stages of infection. Measurements in Acanthamoeba (Supplementary Fig. 2b) showed that the levels of several DUB candidates surpass GroEL, which appears to be lower expressed under amoebal culture conditions. The strongest signals were obtained for SnUSP3, SnCE1, SnVTD, SnJOS1, SnOTU, and SnCE5. Nevertheless, all other mRNAs were also detectable.

### S. negevensis encodes multiple USP-type deubiquitinases with little linkage specificity
Our bioinformatical pipeline will predict not only deubiquitinases but also proteases directed at other ubiquitin-related modifiers ending on

GlyGly. Thus, the first step of our experimental validation strategy measures the reactivity against activity-based probes, in which the last glycine residue of ubiquitin and related modifiers is replaced by a reactive propargyl group (PA)[37]. The three *Simkania* USP-like proteases are large proteins containing hydrophobic regions that might be involved in membrane attachment. Since the full-length proteins could not be expressed in soluble form, experimental validation was performed using the isolated catalytic domains (Fig. 1a). For SnUSP1, we purified the fragment SnUSP1[167–529] and incubated it with a panel of activity-based probes. Reactivity was observed against Ub-PA (Fig. 1b), while no reaction was observed for NEDD8-PA, SUMO1-PA, SUMO3-PA, or ISG15[CTD]-PA (containing the C-terminal ubiquitin-like domain of ISG15). Similarly, SnUSP2[60–401] reacted strongly with Ub-PA and somewhat less with ISG15[CTD]-PA, but not with the other probes (Fig. 1c). SnUSP3[187–522] reacted with Ub-PA and very weakly with ISG15[CTD]-PA (Fig. 1d). A more quantitative comparison of SnUSP1/2/3 activities against the ubiquitin-derived model substrate Ub-AMC is shown in Fig. 1e.

When incubating the catalytic fragment of SnUSP1 (0.5 μM) with 25 μM di-ubiquitin of different linkage types, a rapid and linkage-independent cleavage was observed within 10 min (Fig. 1f). Further dilution of the SnUSP1 fragment (Fig. 1g) revealed subtle activity differences against the individual chain types but confirmed SnUSP1 to be mostly linkage-promiscuous. A similar reactivity profile was observed for the catalytic fragment of SnUSP2 (0.5 μM), with the only difference being that linear di-ubiquitin was cleaved more slowly (Fig. 1h, i). By contrast, the catalytic fragment of SnUSP3 was hardly active against di-ubiquitin species, even when extending the analysis to rare linkage types (Fig. 1j). Only K63, K11 and K6-chains were weakly cleaved after incubation for 3–6 h at a high enzyme concentration (5 μM). Since the same fragment reacted readily with the Ub-PA probe (Fig. 1d), SnUSP3 might be a substrate-directed deubiquitinase rather than a chain-cleaving enzyme. The active site residues of SnUSP1-3 were predicted by homology to eukaryotic USP enzymes (Supplementary Fig. 1a). Replacement of the catalytic cysteine residues by alanine (SnUSP1[C193A], SnUSP2[C101A], and SnUSP3[C209A]) abrogated all reactivity against activity-based probes and ubiquitin chains (Supplementary Fig. 3a–f). Loss of activity was also observed for enzymes mutated in the catalytic histidines SnUSP1[H476A], SnUSP2[H351A], SnUSP3[H472A], or the aromatic gatekeeper residues SnUSP1[Y477A], SnUSP2[Y352A], and SnUSP3[Y473A] (Supplementary Fig. 3a–f). As expected, the Ub-PA probes also reacted with the catalytically inactive DUB versions, as long as the active site cysteine was present[38].

### The first identification of bacterial Josephin-like DUBs

Two Josephin-related open reading frames from *S. negevensis* were identified, which are chromosomal neighbors and share a similar architecture (Fig. 2a). The purified full-length proteins for SnJos1 and SnJos2 did not react with activity-based probes for ubiquitin or ubiquitin-like modifiers NEDD8, SUMO1, SUMO3, and ISG15 (Fig. 2b, c). The mono-ubiquitin based model substrate Ub-AMC was also not cleaved by SnJos1 or SnJos2 (Fig. 2d).

When performing linkage-specific chain-cleaving assays, both enzymes were able to process multiple linkage types. Compared to the USP-DUBs, the *Simkania* Josephins cleave di-ubiquitin chains slowly, requiring several hours at enzyme concentrations of 5 μM (Fig. 2e, f). In this respect, they resemble the slow eukaryotic Josephin DUBs such as ATX3 and JOSD1[39,40]. Both SnJos1 (Fig. 2e) and SnJos2 (Fig. 2f) show the highest activity against K33-linked di-ubiquitin, but also cleave K63, K48, K11, and K6 linkages. In addition, traces of K29 and M1 cleavage were observed with both enzymes. Active site mutants in both enzymes (SnJos1[C212A], SnJos2[C214A], and SnJos2[H345A]) were devoid of chain-cleaving activity, or in the case of SnJos1[H345A] showed a sharp activity reduction (Supplementary Fig. 4a, b). To test whether the poor activity of full-length SnJos1 is caused by auto-inhibitory elements in the non-catalytic region, two truncated versions (SnJos1[171–367] and SnJos1[183–385])

were tested, but found to be inactive (Supplementary Fig. 4c). To further test whether the poor activity was caused by the lack of aromatic gatekeeper motifs, we artificially introduced them as SnJos1[A346W] and SnJos2[A348W]. However, the resulting protein variants turned out to be even less active than the wild type (Supplementary Fig. 4a, b).

### *S. negevensis* encodes a K6-specific deubiquitinase of the VTD-type

The *S. negevensis* genome codes for a single member of the recently established VTD family (Viral Tegument-like Deubiquitinases), which is present in many eukaryotes but also contains two members in the Chlamydia-like bacterium *Waddlia chondrophila*[31]. Since the Simkania VTD enzyme (SnVTD) has an N-terminal membrane-spanning domain (Fig. 3a) and cannot be expressed in soluble form, we analyzed the fragment SnVTD[74–326] for its DUB activity. No reaction with Ub-PA or any other UBL-derived activity-based probes was observed (Fig. 3b). Similarly, the model substrate ubiquitin-AMC was not cleaved (Fig. 3c), suggesting that SnVTD is inactive against mono-ubiquitin substrates. However, the same fragment at a concentration of 0.5 μM showed a strong and specific reactivity with K6-linked di-ubiquitin, leaving all other linkages uncleaved after 3 h (Fig. 3d). By contrast, the active site mutants SnVTD[C104A] and SnVTD[H275A] were inactive in the same assay (Supplementary Fig. 5). Thus, SnVTD shares the activity profile of its *Waddlia chondrophila* homolog WcVTD, which also shows K6-specificity[31].

### The single OTU-type DUB of *Simkania negevensis* is specific for linear chains

*S. negevensis* has one member of the OTU family (SnOTU), which is more similar to eukaryotic otubain proteins than to other OTU sub-families (Supplementary Fig. 1d). In SnOTU, the catalytic domain is followed by a predicted transmembrane region (Fig. 4a). The purified full-length protein was soluble and used for testing reactivity against Ub-PA and related activity-based probes. Only a weak reaction with Ub-PA was observed, while the SUMO-, NEDD8-, and ISG15-based probes did not react (Fig. 4b). The ubiquitin-AMC model substrate was also inert (Fig. 4c), suggesting that SnOTU is inactive against mono-ubiquitin substrates.

When testing the same protein (0.5 μM) on a panel of di-ubiquitin chains of different linkage types, complete cleavage of the M1-linked chains was observed after 10 min, while none of the other chain types showed any cleavage after 60 min (Fig. 4d, Supplementary Fig. 6a). By contrast, the catalytic mutants SnOTU[C82A] and SnOTU[H234A] were inactive against linear di-ubiquitin, and also the gatekeeper mutant SnOTU[H234A] was barely active (Supplementary Fig. 6b). A fragment without the transmembrane domain (SnOTU1-255) showed the same M1-linkage preference as the full-length enzyme (Supplementary Fig. 6c) and had comparable reactivity (Supplementary Fig. 6d).

### Structure of SnOTU with its linear di-ubiquitin substrate

To elucidate the structural basis for M1-specificity of SnOTU, we determined the structure of the catalytically inactivated (C82A) fragment SnOTU[1–255] in complex with linear di-ubiquitin at a resolution of 2.5 Å. Data collection and refinement statistics are given in Table 1. The asymmetric unit contained an unexpected arrangement: Two OTU domains were bridged by two di-ubiquitin molecules, that is, each OTU domain was bound to the second moiety of one di-ubiquitin at its S1 (distal) recognition site, and to the first moiety of another di-ubiquitin at its S1′ (proximal) site (Fig. 5a, Supplementary Fig. 7a). Hence, the N- and C-termini of the two ubiquitin units near each OTU active site are close to each other but are not covalently linked (Supplementary Fig. 7b). Overall, the arrangement resembles the immediate post-cleavage situation, with the two halves of the product still bound to the enzyme. The OTU domain of SnOTU[1–255] is fully resolved from residues 2 to 242 and comprises the OTU-typical papain-fold domain and an

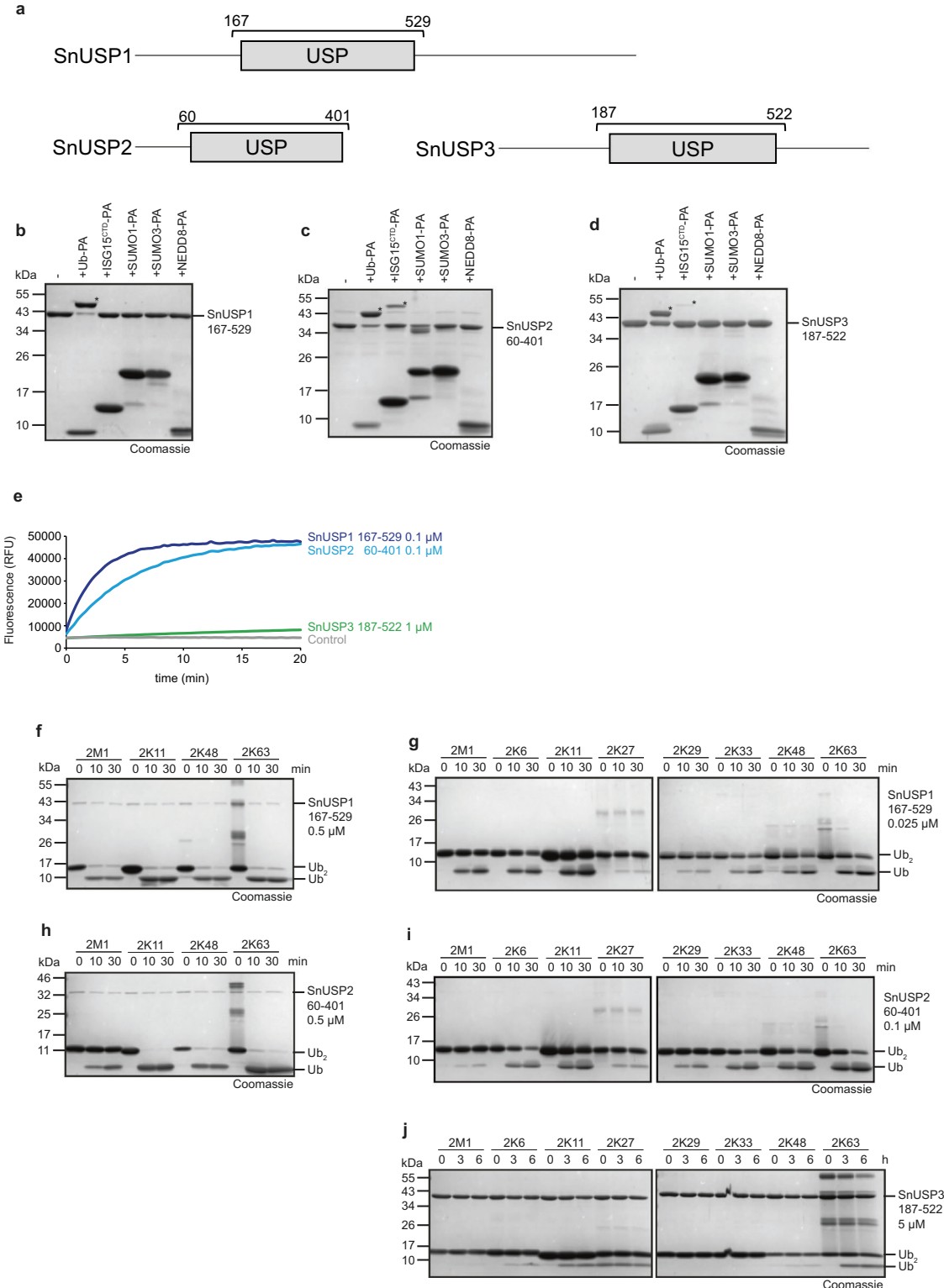

**Fig. 1 | Activity of Simkania USP enzymes. a** Domain scheme of SnUSP1-3. Gray box indicates the DUB domain, bracket indicates the purified construct. **b–d** Activity-based probe reaction of SnUSP1 **b**, SnUSP2 **c**, or SnUSP3 **d** with Ub-PA, ISG15^CTD-PA, SUMO1-PA, SUMO3-PA and NEDD8-PA. Asterisks (*) mark the shifted band after reaction. **e** AMC activity assay with SnUSP1-3 at the indicated concentrations against Ub-AMC substrate shown as released fluorescence (RFU) over time (min). The shown RFU values are the means of triplicates. **f–j** Ubiquitin chain cleavage analysis. A panel of di-ubiquitin chains was treated with 0.5 μM SnUSP1 **f**, 0.025 μM SnUSP1 **g**, 0.5 μM SnUSP2 **h**, 0.1 μM SnUSP2 **i**, or 5 μM SnUSP3 **j** for the indicated time points. Source data are provided as a Source Data file.

additional N-terminal β-hairpin motif consisting of strands β1 and β2 (Fig. 5b). In agreement with the OTU alignment (Supplementary Fig. 1d), the active site is formed by Cys-82 and His-234. Leu-236 is positioned where several other OTUs have a third active site residue[25],

but it is not able to polarize His-234 and is thus not expected to be catalytically important (Fig. 5c). The importance of an acidic third active site residue in the OTU family is known to vary: in some viral OTUs, this acidic residue is completely absent[41] while in several human

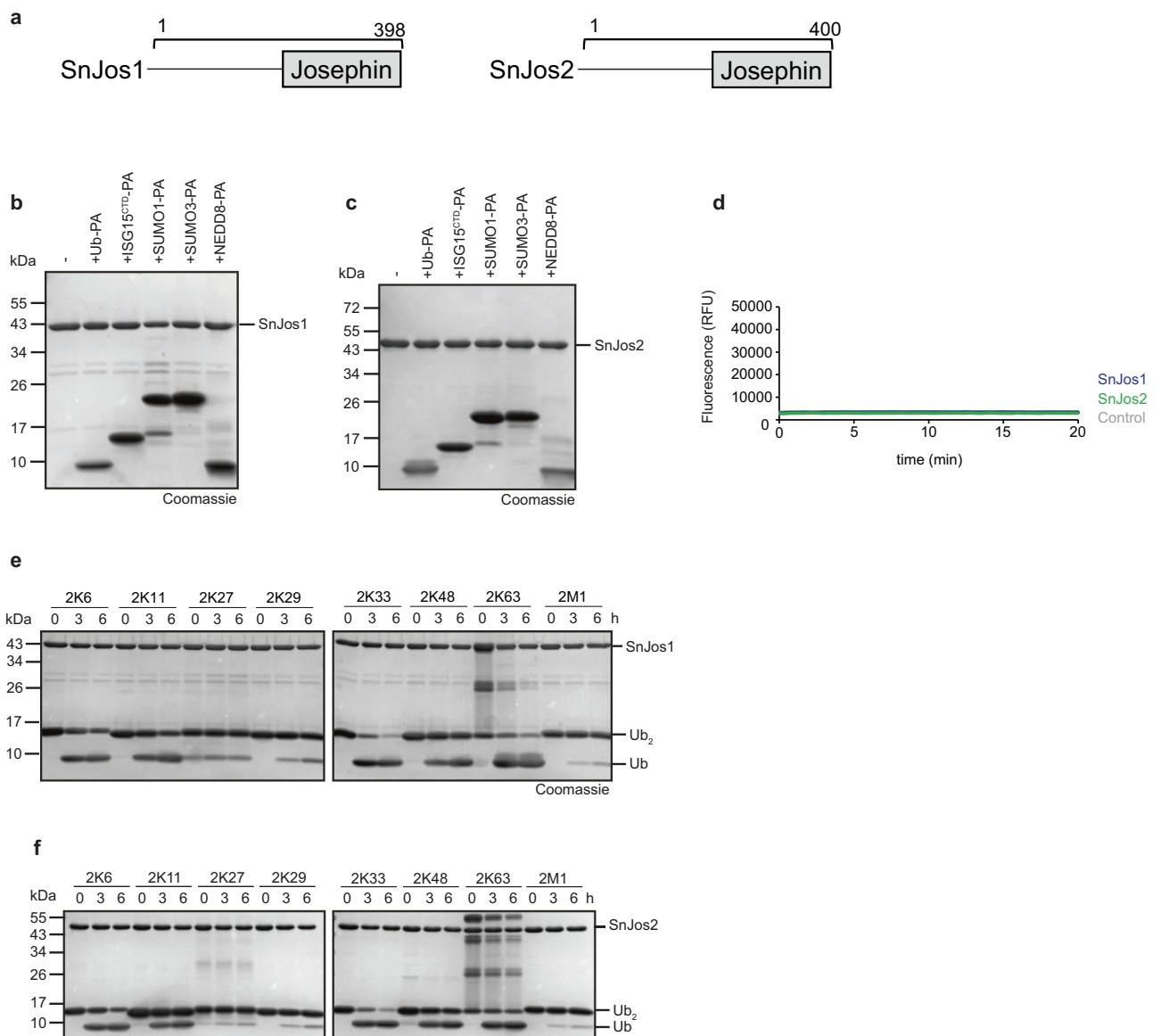

**Fig. 2 | Activity of *Simkania* Josephin-type enzymes. a** Domain scheme of SnJos1-2. Gray box indicates the DUB domain, bracket indicates the purified construct. **b**, **c** Activity-based probe reaction of SnJos1 **b** or SnJos2 **c** with Ub-PA, ISG15^CTD-PA, SUMO1-PA, SUMO3-PA, and NEDD8-PA. **d** AMC activity assay with SnJos1-2 against the Ub-AMC substrate shown as released fluorescence (RFU) over time (min). The shown RFU values are the means of triplicates. **e**–**f** Ubiquitin chain cleavage analysis. A panel of diubiquitin chains was treated with 5 μM SnJos1 **d**, or 5 μM SnJos2 **e** for the indicated time points. Source data are provided as a Source Data file.

and bacterial OTU-type DUBs, an acidic third residue is present but can be mutated without much impact on catalysis[25]. The unusual arrangement of the M1 di-ubiquitin in the experimentally determined complex structure (PDB:8CMR) leads to unintuitive residue numbering, where residues 1–76 correspond to the S1′-bound ubiquitin and residues 77–152 to the S1-bound ubiquitin (Supplementary Fig. 7a). To avoid confusion while discussing the contact residues and surface patches, canonical ubiquitin numbering 1–76 will be used for both S1 and S1′-bound moieties.

The poor activity of SnOTU against Ub-PA suggests an important contribution of the S1′ ubiquitin to the overall substrate recognition. Indeed, the proximal ubiquitin is bound by an extensive surface with major contributions from the N-terminal β1/β2-hairpin motif (Fig. 5b); deletion of this region leads to complete inactivity (Supplementary Fig. 7c). Tyr-13 and Phe-15 within the β1-strand and Tyr-20 located in strand β2 make hydrophobic interactions with the Ile-44 patch of the S1′-ubiquitin (Ile-44, His-68, Val-70) (Fig. 5d). In

addition, Glu-16 at the β1/β2-turn forms a salt bridge with Arg-42 of ubiquitin (Fig. 5d). Contacts outside the β1/β2-hairpin also contribute to the S1′ recognition surface: Arg-230 and Lys-231 form salt bridges with Glu-18 and Glu-16 of S1′-ubiquitin, respectively (Fig. 5e), whereas Phe-25 interacts with Phe-4 of ubiquitin (Supplementary Fig. 7d). Individual Ala substitutions were generated for each of these residues and showed a loss of activity against linear ubiquitin chains (Fig. 5f, Supplementary Fig. 7e). The importance of the two salt bridges of SnOTU (Arg-230 and Lys-231 with Glu-18 and Glu-16 of S1′-ubiquitin) was confirmed by two mutants in the M1-linked di-ubiquitin substrate: Both E16A(S1′) and E18A(S1′) are much poorer substrates than wild-type di-ubiquitin (Fig. 5g). The contact of His-49 with Asp-34 of S1′-ubiquitin (Fig. 5e) appears less crucial, as the H49A mutant retains some catalytic activity (Fig. 5f). The Met-1 residue of S1′-ubiquitin is not involved in critical contacts, and the M1A(S1′) mutation showed the same cleavage behavior as wild-type di-ubiquitin (Supplementary Fig. 7f).

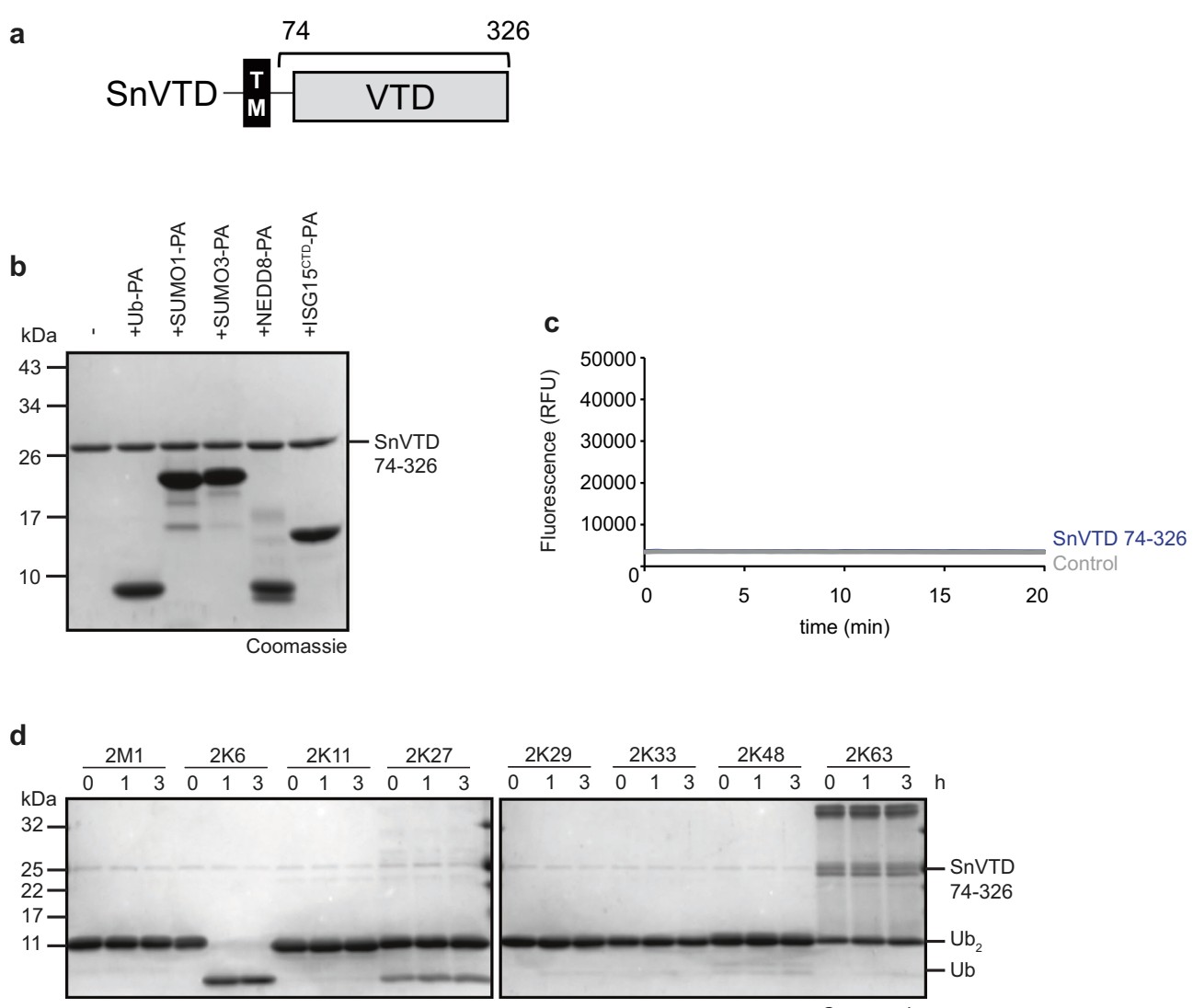

**Fig. 3 | _Simkania_ encodes a K6-specific VTD-type enzyme. a** Domain scheme of SnVTD. The gray box indicates the DUB domain, the black box marks a putative transmembrane region, and the bracket indicates the purified construct. **b** Activity-based probe reaction of SnVTD with Ub-PA, ISG15$^{CTD}$-PA, SUMO1-PA, SUMO3-PA, and NEDD8-PA. **c** AMC activity assay with SnVTD against Ub-AMC substrate is shown as released fluorescence (RFU) over time (min). The shown RFU values are the means of triplicates. **d** Ubiquitin chain cleavage analysis. A panel of di-ubiquitin chains was treated with 0.5 μM SnVTD for the indicated time points. Source data are provided as a Source Data file.

The distal (S1) ubiquitin shows a less extensive contact network than the proximal ubiquitin, and mutations in the S1-contacting residues tend to have milder effects on M1 cleavage activity. Most deubiquitinases tether the C-terminal portion of the S1 ubiquitin by forming salt bridges with the arginine residues of its RLRGG motif. In the SnOTU structure, Glu-183 interacts with Arg-74 and the amino-terminus of the S1-ubiquitin (Fig. 5h), and a salt bridge appears between SnOTU Glu-185 and Arg-42 of the S1-ubiquitin (Fig. 5i). Surprisingly, neither the E183A mutant of SnOTU nor the R74A mutant of S1 ubiquitin had a strong effect (Fig. 5j, k), suggesting that this salt bridge is not crucial for activity. By contrast, the salt bridge between SnOTU Glu-185 and Arg-42 of the S1-ubiquitin is crucial, as E185A is the only S1-recognition mutant that completely abrogates activity (Fig. 5j). Arg-72 of the S1 ubiquitin does not form a salt bridge in the structure (Fig. 5h) but seems to be important for ubiquitin recognition, as the R72A(S1) mutant is a poor substrate (Fig. 5k). Alanine mutations of several other S1-contacting residues lead to reduced catalytic activity, as seen for Glu-174 (contacting S1 Arg-42), Phe-168 and Leu-171 (both

contacting the S1-ubiquitin Ile-44 patch), and Tyr-204 (contacting the S1-ubiquitin Ile-36 patch) (Fig. 5h, i, j).

The crucial importance of S1′-ubiquitin and comparative lack of S1-specificity is highlighted by the different reactivities of SnOTU towards M1-linked mixed chains. A Ub-Nedd8 fusion protein, which is efficiently processed by the S1-recognizing DUB USP21, is not cleaved at all by SnOTU (Supplementary Fig. 7g). By contrast, the Nedd8-Ub fusion, which contains the crucial S1′-ubiquitin, is readily cleaved by SnOTU, but not by USP21 (Supplementary Fig. 7h).

## _S. negevensis_ encodes multiple CE-clan proteases, one of which has deSUMOylase and DUB activities

Overall, five _S. negevensis_ open reading frames with similarity to CE-clan enzymes were detected (Fig. 6a). The first member, SnCE1 (SNE_A10940), did not react with the activity-based probe Ub-PA (Fig. 6b) and showed weak activity against the fluorogenic model substrate Ub-AMC (Fig. 6c). SnCE1 at 5 μM concentration was able to slowly cleave K11, K48, and K63 di-ubiquitin, but was inactive against

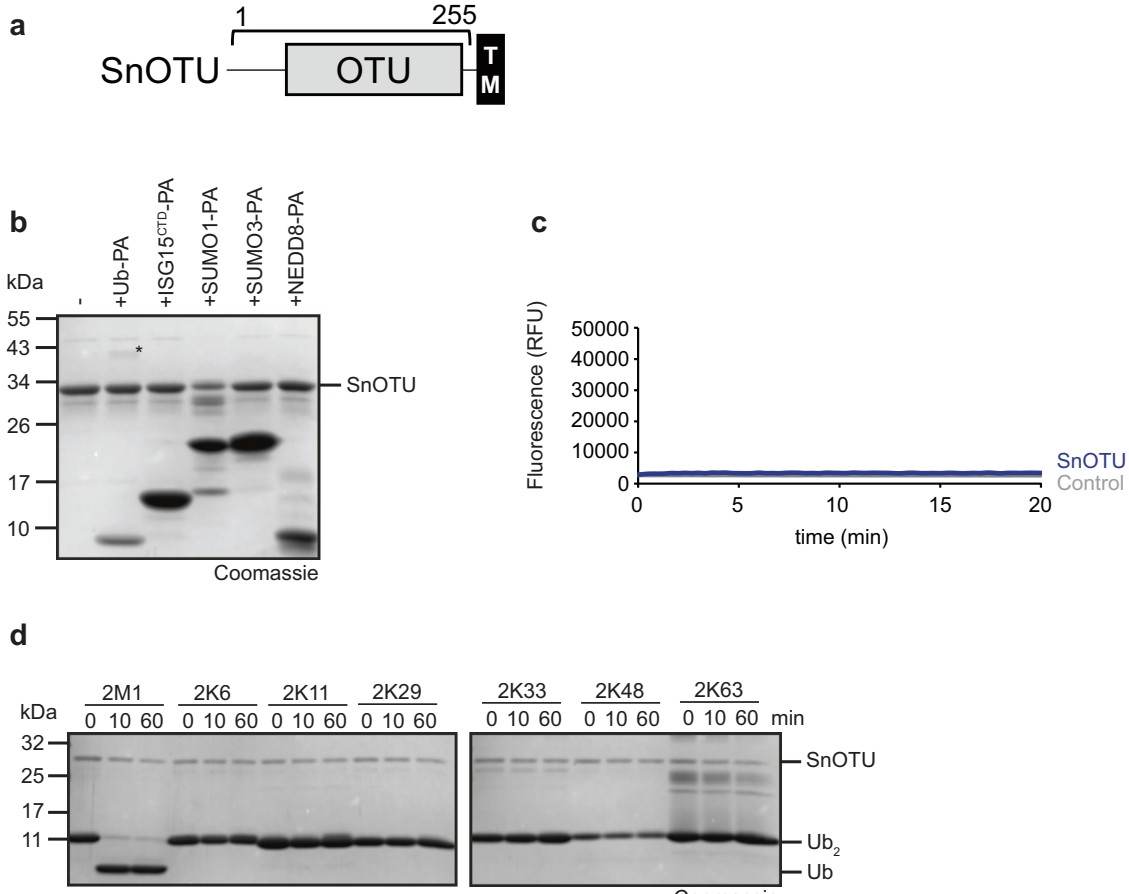

**Fig. 4 | *Simkania* encodes an M1-specific OTU-type enzyme. a** Domain scheme of SnOTU. The gray box indicates the DUB domain, the black box marks a putative transmembrane region, and the bracket indicates the purified construct. **b** Activity-based probe reaction of SnOTU with Ub-PA, ISG5$^{CTD}$-PA, SUMO1-PA, SUMO3-PA, and NEDD8-PA. **c** AMC activity assay with SnOTU against Ub-AMC substrate is shown as released fluorescence (RFU) over time (min). The shown RFU values are the means of triplicate measurements. **d** Ubiquitin chain cleavage analysis. A panel of di-ubiquitin chains was treated with 0.5 μM SnOTU for the indicated time points. Source data are provided as a Source Data file.

other chains (Fig. 6d). The catalytic mutants SnCE1$^{C256A}$ and SnCE1$^{H190A}$ were inactive, and the same was true for the gatekeeper mutant SnCE1$^{W191A}$ (Supplementary Fig. 8a). Similar to what had been observed for the SnUSP enzymes, the Ub-PA probe still reacted with the mutants SnCE1$^{H190A}$ and SnCE1$^{W191A}$, but required the presence of the catalytic cysteine (Supplementary Fig. 8b).

Unlike all other *S. negevensis* DUBs, SnCE1 reacted readily with the SUMO-based probes SUMO1-PA and SUMO3-PA (Fig. 6b), was active against SUMO1-AMC and SUMO2-AMC model substrates (Fig. 6c), and also against authentic SUMO2 chains (Fig. 6e). At 2.5 μM enzyme concentration, all high-MW SUMO2 chains were cleaved within 10 min. Both the direct comparison of AMC-substrate activities and chain cleavage results showed that SnCE1 is more active as a deSUMOylase than as a deubiquitinase. The other four *Simkania* members of the CE-clan neither cleaved any of the activity-based probes (Supplementary Fig. 8c–f), nor were they active against ubiquitin chains (Supplementary Fig. 8g–j) and thus were not further studied in detail.

## Discussion

To survive and proliferate inside the host cell, intracellular bacteria must create niches within the cell and interfere with the host cell defense system. In the ongoing arms race between pathogen and host cells, the development of new strategies to survive intracellularly is of the greatest importance. The host-encoded ubiquitin system is both a threat and opportunity for intracellular pathogens. On the one hand, ubiquitination by host ligases is directed to the surface components of

bacteria and their surrounding vacuoles, targeting the bacteria for lysosomal degradation[5]. On the other hand, bacteria can use their own ligase effectors to divert the ubiquitin system and target antibacterial host factors[42]. Since the ubiquitin system is highly conserved in all eukaryotes, ubiquitin-directed effectors are found in pathogens of humans, animals, and plants[12]. A large number of deubiquitinase (DUB) effectors of intracellular bacteria have been described, which are thought to keep the bacterial surface and bacteria-containing vacuoles free from ubiquitin modifications. With the exception of *L. pneumophila*, bacterial DUBs have no particular ubiquitin linkage specificity; they often prefer K63-linked chains, which are a typical signal for autophagy and vesicular trafficking[43,44].

With very few exceptions, known bacterial DUBs belong to two classes of cysteine proteases: the OTU and CE families. So far, DUB effectors have mainly been studied in human pathogens belonging to a single bacterial phylum that includes Alphaproteobacteria (*Orientia*, *Rickettsia*), Betaproteobacteria (*Burkholderia*), and Gammaproteobacteria (*Salmonella*, *Shigella*, *Legionella*). Knowledge about DUBs of other bacterial phyla is currently restricted to *C. trachomatis* and *C. pneumoniae*, which encode two CE-type DUBs or one OTU-type DUB, respectively[17,18]. Surprisingly, the *Chlamydia*-like bacterium *S. negevensis* possesses no less than eight active deubiquitinases and four related sequences without detectable DUB activity. Thus, the size of the *S. negevensis* DUB repertoire resembles that of *L. pneumophila*, whereas no more than two DUBs have been identified in any other bacterium. Another feature shared by *S. negevensis* and *L.*

**Table 1 | Data collection and refinement statistics**

| Structure name | SnOTU (PDB entry 8CMR) |
|---|---|
| Wavelength (Å) | 1.00 |
| Resolution range (Å) | 43.06–2.24 (2.32–2.24) |
| Space group | P 61 |
| Unit cell (Å) | 154.618 154.618 81.972 90 90 120 |
| Total reflections | 603643 (62262) |
| Unique reflections | 53772 (5336) |
| Multiplicity | 11.2 (11.7) |
| Completeness (%) | 99.92 (99.64) |
| Mean I/sigma(I) | 8.26 (1.09) |
| Wilson B-factor (Å²) | 57.20 |
| R-merge | 0.135 (1.972) |
| R-meas | 0.141 (2.062) |
| R-pim | 0.042 (0.6007) |
| CC1/2 | 0.997 (0.603) |
| CC* | 0.999 (0.867) |
| Reflections used in refinement | 53730 (5319) |
| Reflections used for R-free | 1339 (135) |
| R-work | 0.1945 (0.3374) |
| R-free | 0.2131 (0.4100) |
| Number of non-hydrogen atoms | 6300 |
| macromolecules | 6204 |
| ligands | 0 |
| solvent | 96 |
| Protein residues | 783 |
| RMS(bonds) (Å) | 0.001 |
| RMS(angles) (degrees) | 0.52 |
| Ramachandran favored/allowed/outliers (%) | 97.41/2.59/0.00 |
| Rotamer outliers (%) | 0.87 |
| Clashscore | 0.56 |
| Average B-factor (Å²) | 65.88 |
| macromolecules | 65.99 |
| solvent | 58.51 |

Statistics for the highest-resolution shell are shown in parentheses.

*pneumophila*, but missing from other bacteria, is the presence of highly linkage-specific DUBs for linear (M1) chains and for K6-linked chains. While there is a functional analogy between *Simkania* and *Legionella* DUB repertoires, their member enzymes are not homologous and *S. negevensis* shows an overall higher DUB class diversity. *L. pneumophila* encodes—apart from the idiosyncratic RavD—exclusively DUBs of the OTU and CE families. By contrast, *S. negevensis* uses OTU and CE-type enzymes in combination with DUB classes not seen in other bacteria, including USPs, Josephins, and VTD-type deubiquitinases.

Why *L. pneumophila* and *S. negevensi*s, two unrelated bacteria belonging to different phyla, share such extensive and functionally analogous sets of deubiquitinase effectors can only be speculated. *L. pneumophila* occurs naturally in fresh water amoebae, but can—when taken up via aerosol—also infect human alveolar macrophages and cause Legionnaire's disease[45]. *S. negevensis* also grows in amoebae and can infect a wide range of hosts, including human macrophages[46,47]. *S. negevensis* is highly seroprevalent in several human populations and has been suggested to cause pneumonia; however, its overall contribution to this disease is considered marginal[28]. The similarly broad host ranges of *S. negevensis* and *L. pneumophila* might have selected

for their expanded DUB repertoire. Little is known about the antibacterial defense mechanisms of unicellular eukaryotes[48,49], but it can be assumed that ubiquitin and ubiquitin-dependent autophagy play an important role. Considering the extreme conservation of ubiquitin, effector deubiquitinases are functional in most eukaryotes. At the same time, pathogen evolution should favor the acquisition of DUBs that do not interfere with the bacterially encoded E3 ligases. One possibility for avoiding this conflict is a separation of the linkage specificities, for example, through the inactivity of bacterial DUBs towards K48 chains generated by the bacterial ligases that target host proteins for destruction. An alternative, but not mutually exclusive strategy is based on the specific localization of DUB effectors to the bacterial surface or bacteria-containing vacuoles, while avoiding the sites of bacterial E3s activity. As different host organisms may use different ligases and linkage types to initiate xenophagy, a diversified DUB repertoire would be advantageous for bacteria with wide host ranges.

The current inaccessibility of *S. negevensis* for genetic manipulations precludes experimental verification of DUB secretion into the host cell cytoplasm. *Simkania* possesses a type-III secretion system and a plasmid-encoded type-IV secretion system[50,51] able to secrete effectors into eukaryotic host cells. However, the prediction of type-III and type-IV secretion signals is not straightforward[52]. Among the DUB candidates described here, SnUSP2, SnJOS1, SnJOS2, SnCE2, and SnCE4 are predicted to have type-III secretion signals[52], while SnVTD and SnOTU contain hydrophobic transmembrane helices. Besides recognizable secretion signals, a relationship to typically eukaryotic protein families is often used for effector prediction, for example, by the EffectiveELD predictor used by EffectiveDB[52]. Deubiquitinase domains belong to the typical eukaryotic protein families and are also abundant in known bacterial effectors, supporting the idea that the Simkania DUBs are secreted into a eukaryotic host—but might be host-selective.

The specific requirement for K6- and M1-linked DUBs is difficult to explain by the host range alone. K6-linked ubiquitin chains have been implicated in a wide variety of biological pathways[53], but only one occurrence of antibacterial K6-chains assembled by the ligase LRSAM1 has been reported[54]. It is possible that K6 chains are involved in non-mammalian defense systems; however, data supporting this idea are currently lacking. K6 chains are also important for the regulation of mitophagy[53]. Both *L. pneumophila* and *S. negevensis* have been shown to modulate host cell mitochondrial dynamics to the advantage of the pathogen, including induction of extensive mitochondrial fission[29,55]. Thus, it is conceivable that these bacteria interfere with mitophagy by hydrolyzing the K6-linked ubiquitin chains.

Linear (M1) chains are known to restrict intracellular bacteria in human cells by targeting substrates for xenophagy, as shown for *Salmonella*[56,57] and *Legionella*[22]. However, linear chains are considered a metazoan invention, as only animals encode the subunits of the LUBAC complex, the only known M1-specific ligase[58]. Thus, M1-specific DUBs can hardly be explained by promiscuity for non-metazoan hosts. A possible explanation for the lack of M1-specific DUBs in other bacteria is the use of alternative mechanisms to avoid decoration by linear chains. Recently, the effector protein GarD has been shown to shield *C. trachomatis* from linear chains, not by deubiquitination, but by altering the inclusion membrane to restrict access by ubiquitin ligases[11]. Homologs of GarD are present in other *Chlamydiae* but not in *S. negevensis*, compatible with the idea of GarD and SnOTU as alternative strategies for avoiding M1 ubiquitination. Since HOIP1, the M1-specific ligase of the LUBAC complex, does not modify the substrate itself, but rather a substrate-bound ubiquitin or ubiquitin chain[58,59], bacteria might use DUBs of other specificities to target the substrate-proximal ubiquitin and thus indirectly remove M1-chains.

Another factor that potentially demands a diversified DUB repertoire is the unusually protracted lifecycle of *S. negevensis*. In contrast to the infection cycle of *Chlamydia*, the development of *S.*

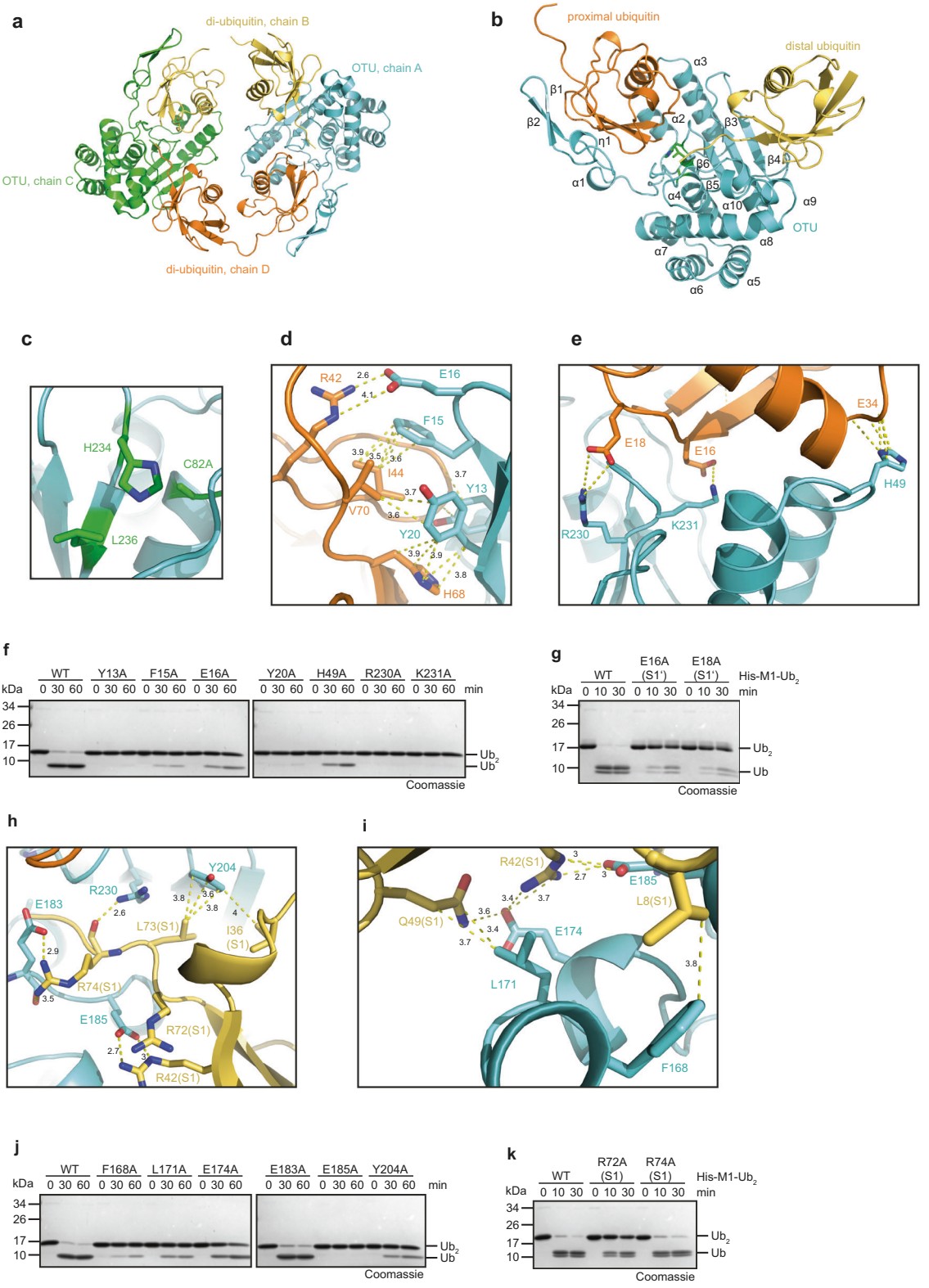

*negevensis* takes up to 12–15 days, depending on the cell line. During this time, the number of bacterial particles increases considerably without killing the host cell and spreading the particles to uninfected neighboring cells. On day 2–4 pi, the bacterium reaches its replication plateau, and the number of reticulate bodies (RBs) within the *Simkania*-containing vacuole reaches the maximum[60]. After re-differentiation into elementary bodies (EBs), the bacteria exit the host cell through a mechanism that involves caspases, especially

caspase 1, as well as myosin II[30]. A prolonged presence inside viable cells increases exposure to intracellular defense mechanisms and the risk of becoming subject to ubiquitination-induced xenophagy.

Apart from their physiological importance, the diversified DUB repertoire of *S. negevensis* is also interesting with regard to DUB evolution and specificity. First, it should be emphasized that M1 and K6 specific activities in *Legionella* and *Simkania* arose through convergent evolution. The K6-specific DUB of *L. pneumophila* (LotA)[23,24]

**Fig. 5 | Crystal structure of SnOTU in complex with linear di-ubiquitin.**
**a** Content of the asymmetric unit with two SnOTU molecules bridged by two M1-linked di-ubiquitin chains. There are two SnOTU and two di-ubiquitin molecules, which are shown in cartoon representation and colored green/teal or yellow/orange, respectively. The residues connecting the two ubiquitin moieties in the di-ubiquitin molecules are disordered in chain B, but well defined in the electron density in chain D. **b** Overview of the SnOTU/di-ubiquitin co-structure in cartoon representation. The catalytic domain is colored teal. The proximal (orange) and distal (yellow) ubiquitin originate from two distinct linear di-ubiquitins and resemble a post-cleavage complex. **c** Magnified view of the SnOTU active site. Cys-82 was replaced by Ala in the present structure and forms the active site together with His-234. Leu-236 occupies the position of the third active site residue but is unable to polarize His-234. **d, e** Intensive interactions between the Ile-44 patch of S1' ubiquitin (colored orange) and the N-terminal β1/β2-hairpin motif **d**, or the catalytic core domain **e** of SnOTU (colored teal). Residues involved in these interactions are

shown as sticks. Interactions are indicated by yellow dotted lines. **f** Activity of S1' site mutants against linear linked di-ubiquitin. Linear di-ubiquitin was incubated with 0.1 μM wildtype SnOTU or the indicated mutants. **g** Activity of wild-type SnOTU against ubiquitin mutants. N-terminally His-tagged and mutated linear di-ubiquitin was incubated with 0.25 μM SnOTU for the indicated timepoints. **h–i** Recognition of the distal ubiquitin by SnOTU (colored teal) at the ubiquitin C-terminus region **h** and through additional interactions **i**. The S1 bound ubiquitin is colored yellow. Residues involved in these interactions are shown as sticks. Interactions are indicated by yellow dotted lines; the numbers represent the distances in Å. **j** Activity of S1 site mutants against linear-linked diubiquitin. Linear linked di-ubiquitin was incubated with 0.1 μM wildtype SnOTU or the indicated mutants. **k** Activity of wild-type SnOTU against ubiquitin mutants. N-terminal His-tagged and mutated linear linked di-ubiquitin (R72/R74 of distal ubiquitin) was incubated with 0.25 μM SnOTU for the indicated timepoints.

belongs to the OTU family, which is known for comprising mostly linkage-specific or -selective activities[61]. By contrast, the K6-specific DUB of *S. negevensis* (SnVTD) belongs to the recently discovered VTD family, which is unrelated to OTUs but fulfills similar roles in viruses and transposons[31]. One structural feature shared by these two dissimilar enzymes is the presence of a loop interacting with the Ile-36 surface patch of ubiquitin, which might be a defining feature for K6-specificity. Apart from LotA, SnVTD, and WcVTD1, no other strictly K6-specific deubiquitinases are known, which makes these bacterial DUBs valuable tools for general studies on K6-linked ubiquitin chains by UbiCRest-type experiments[61].

RavD, the M1-specific DUB of *L. pneumophila*, is the only bacterial deubiquitinase that cannot be assigned to a eukaryotic DUB family, although it assumes the DUB-typical papain fold[22]. By contrast, the M1-specific SnOTU from *S. negevensis* belongs to the OTU family, which also contains OTULIN, the only known eukaryotic M1-specific DUB[32]. However, SnOTU is not a bacterial version of OTULIN, but rather belongs to the OTUB subfamily, whose eukaryotic members have different linkage preferences. It must therefore be assumed that *S. negevensis* or its ancestor co-opted a host-derived OTUB enzyme and changed its linkage specificity to meet the bacterium's need for M1 cleavage. The different evolutionary origins of OTULIN and SnOTU are also reflected by their different mechanisms for achieving M1-specificity. OTULIN uses substrate-assisted catalysis, requiring an S1'-bound ubiquitin in M1-linkage to release the active site His-339 residue from being tethered in an unproductive conformation[32]; a similar mechanism has been proposed for RavD[62]. SnOTU achieves M1-specificity by a particularly extensive recognition surface for the M1-linked ubiquitin at the S1' position (Fig. 5). Unlike typical DUBs, SnOTU is rather resilient to mutations in the S1 recognition surface, whereas mutations in the S1' recognition surface abrogate all DUB activity. This unusual specificity is highlighted by the inactivity of SnOTU towards a linear Ub-Nedd8 fusion, while the Nedd8-Ub fusion is efficiently cleaved (Supplementary Fig. 7f, g).

The importance of S1' recognition for the M1-specificity by OTU deubiquitinases is also corroborated by a structural superposition of SnOTU and human OTULIN, with and without bound di-ubiquitin (Supplementary Fig. 9b–d). The conformation of the catalytic triad in its productive ubiquitin-bound form is similar (Supplementary Fig. 9c) and shows that Leu-236 of SnOTU assumes the position of the third catalytic residue Asn-341 of OTULIN. Interestingly, the proximal S1'-ubiquitin shares the same orientation relative to the SnOTU and OTULIN active residues, while the S1 ubiquitin is slightly rotated (Supplementary Fig. 9d). In both structures, the catalytic histidine is contacted by Glu-16 of the S1'-ubiquitin—the residue that is crucial for substrate-assisted catalysis of OTULIN[32] and is also important for SnOTU activity (Fig. 5g).

The presence of USP and Josephin-type deubiquitinases in *S. negevensis* shows that bacteria can also co-opt DUB types other than

OTU and CE-clan enzymes, and suggests that the skew of published DUBs towards these classes is partially due to observational bias. The *Simkania* USP enzymes SnUSP1 and SnUSP2 are highly active and linkage-promiscuous, whereas SnUSP3 reacts with ubiquitin model substrates but does not cleave chains. Both these findings are in agreement with known eukaryotic USP activities, which often lack linkage-selectivity and can be directed to particular substrates[63]. While SnUSP1 appears to be ubiquitin-specific, SnUSP2 also reacts with an ISG15-derived activity-based probe (Fig. 1b, c). In general, bacterial ISG15-directed effectors are rare, but the SnUSP1 activity might help *Simkania* to evade the interferon-based innate immunity, which is known to be active against *Chlamydiales* bacteria[64].

The two Josephin-type DUBs, SnJos1, and SnJos2, show modest chain-cleaving activity and little linkage specificity. These properties are in line with eukaryotic Josephins, which are also slow to cleave standard chain types and have been proposed to act on unusual substrates, such as mixed chains[40] or ester-linked ubiquitin[65]. Some mammalian Josephins are activated by posttranslational modifications —a regulation mode that might also work with bacterial DUB effectors[39].

Among the five *Simkania* ORFs with CE-clan similarities, only SnCE1 showed detectable DUB activity. This enzyme is unusual in that it also has strong deSUMOylase activity, which is rarely observed in bacteria. One known example of a CE-clan enzyme with dual DUB/deSUMOylase activity is XopD from *Xanthomonas campestris*, a plant pathogen[26,66]. For animals, a role of SUMOylation in antibacterial defense has not been described, suggesting that the deSUMOylating role of SnCE1 might have evolved for coping with non-metazoan hosts. The other four SnCE proteins lack DUB and deSUMOylase activities, which is not unusual for CE-clan enzymes, which also include acetyl-transferases and other proteases[26]. In summary, our study shows that the opportunistic pathogen *Simkania negevensis* encodes a large and diverse repertoire of deubiquitinases, which functionally—but not structurally—resembles that of *Legionella pneumophila* and has been acquired independently. Our data suggest that the scope of ubiquitin-directed effectors is shaped by lifestyle and host range rather than by common ancestry.

## Methods
### Sequence analysis
All sequence alignments were generated using the MAFFT package[67]. Generalized profiles were calculated from multiple alignments using pftools[33] and searched against all proteins from the Uniprot database (https://www.uniprot.org). HMM-to-HMM searches were performed using the HHSEARCH method[34]. The transmembrane regions were predicted using TMHMM v2.0[68]. Structural comparisons were performed using the DALI server[69]. AlphaFold v2.2[70] predictions were carried out on the HPC system of the University of Cologne, the Cologne High-Efficient Operating Platform for Science (CHEOPS).

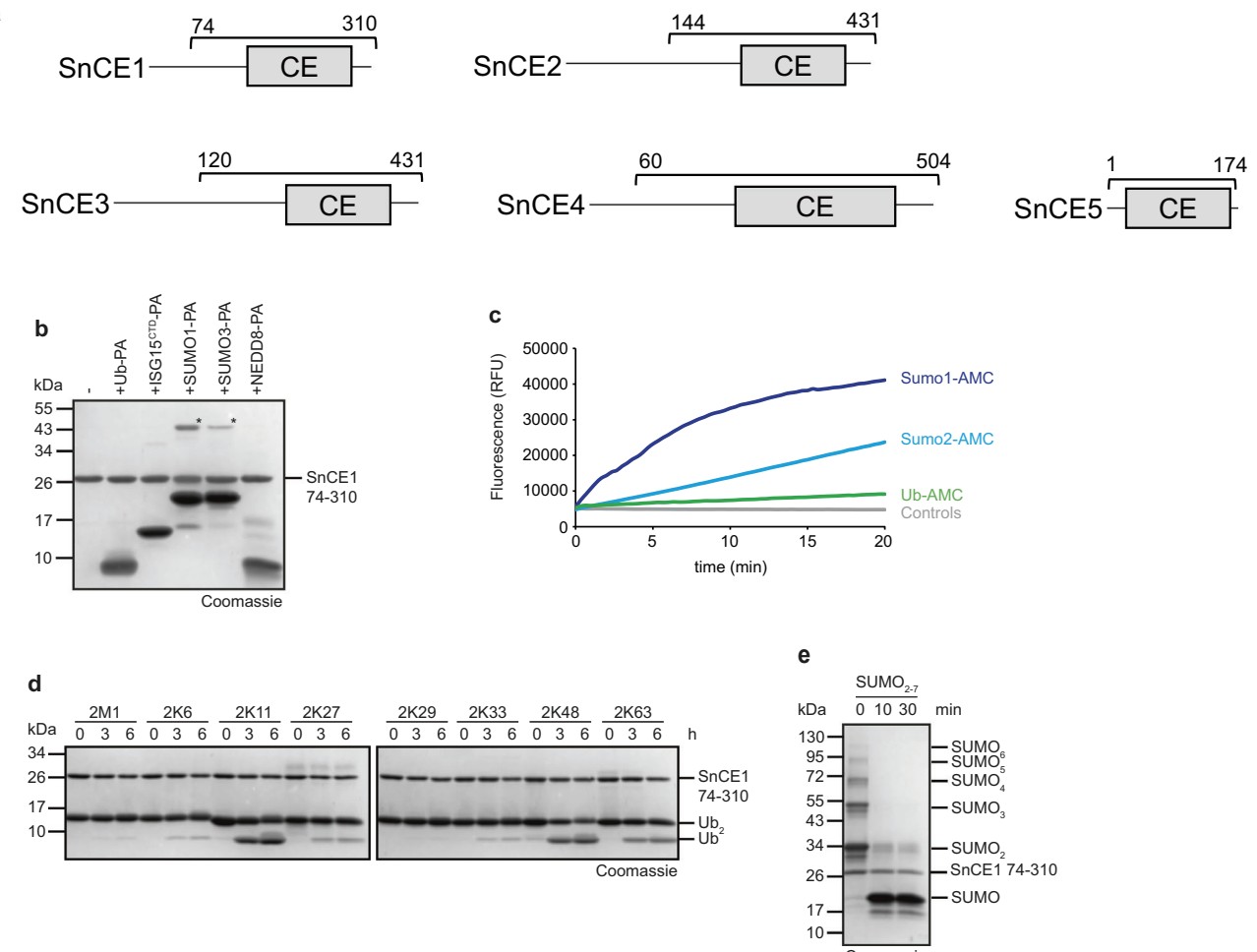

**Fig. 6 | Activities of *Simkania* CE1. a** Domain scheme of SnCE1-5. The gray box indicates the DUB domain and the bracket indicates the purified construct. **b** Activity-based probe reaction of SnCE1 with Ub-PA, ISG15$^{CTD}$-PA, SUMO1-PA, SUMO3-PA, and NEDD8-PA. **c** AMC activity assay with SnCE1 against Ub-/SUMO1-/SUMO2-AMC substrates is shown as released fluorescence (RFU) over time (min). The shown RFU values are the means of triplicates. **d** Ubiquitin chain cleavage analysis. A panel of diubiquitin chains was treated with 5 μM SnCE1 at the indicated time points. **e** Activity assay with SnCE1 against SUMO chains. A mixture of different SUMO chain lengths was treated with the DUB candidate at the indicated time points. Source data are provided as a Source Data file.

## Cloning & mutagenesis

The SnJos1 coding region was obtained by gene synthesis (IDT). The SnUSP3 coding region was amplified from *S. negevensis* genomic DNA. Coding regions from all other DUB candidates were amplified from *S. negevensis* genomic DNA (ordered from the Leibniz Institute DSMZ, German Collection of Microorganisms and Cell Cultures, DSM No.27360). All amplifications were performed by PCR using the Phusion High Fidelity Kit (New England Biolabs). The PCR fragments and gBlocks were cloned into pOPIN-S and pOPIN-K vectors[71] using the In-Fusion HD Cloning Kit (Takara Clontech). Point mutations were introduced using the QuikChange Lightning kit (Agilent Technologies). A list of primers is provided in the Supplementary Table 1.

Constructs for ubiquitin-PA purification (pTXB1-ubiquitin1–75) were a kind gift from David Komander (WEHI, Melbourne). SUMO11–96, SUMO31–91, and ISG1579–156 were amplified by PCR with an N-terminal 3xFlag tag and cloned into the pTXB1 vector (New England Biolabs) by restriction cloning according to the manufacturer's protocol.

## Infection and qPRC analysis

HeLa 229 cells (ATCC, CCL-2.1) were grown in RPMI medium supplemented with 10% FCS at 5% $CO_2$/37 °C. Cells were infected with *S. negevensis* at an estimated MOI 1 in the RPMI medium containing 5%

FCS and incubated at 5% $CO_2$/35 °C until the end of experiment. Samples were collected on days 2 and 4 post infection, and quantitative PCR was performed. A list of primers is provided in the Supplementary Data. ΔCt values were calculated in comparison to the host cell housekeeping gene *GAPDH* for all genes, and ΔΔCt values were calculated as mRNA amount for individual genes in comparison to the non-infected control.

Amoeba (*Acanthamoeba castellanii*) stably infected with *Simkania negevensis* were grown in cell culture dishes in HL5 rich axenic medium (56 mM glucose, 2.6 mM KH2PO4, 1.3 mM Na2HPO4x7H2O, 0.5% proteose peptone 2, 0.5% thiotone E peptone, 0.5% yeast extract) at 30 °C. After mRNA isolation and cDNA preparation, qPCR was performed. A list of primers is provided in the Supplementary Data. ΔCt values were calculated in comparison to the host cell housekeeping gene 5 S RNA (18SQV and HPRT were also measured to monitor reliability[72]) for all genes, and ΔΔCt values were calculated as mRNA amount for individual genes in comparison to the non-infected control.

## Protein expression & purification

SnVTD and SnCE1 were expressed from the pOPIN-K vector with an N-terminal 6His-GST-tag, and all other DUB candidates were expressed from the pOPIN-S vector with an N-terminal 6His-Smt3-tag. *Escherichia coli* (Strain: Rosetta (DE3) pLysS) were transformed with constructs

expressing DUBs and 2–6 l cultures were grown in LB medium at 37 °C until an $OD_{600}$ of 0.8 was reached. The cultures were cooled down to 18 °C and protein expression was induced by addition of 0.1 mM iso-propyl β-d-1-thiogalactopyranoside (IPTG).

After 16 h, the cultures were harvested by centrifugation at $5000 \times g$ for 15 min. After freeze-thawing, the pellets were resuspended in binding buffer (300 mM NaCl, 20 mM TRIS pH 7.5, 20 mM imidazole, 2 mM β-mercaptoethanol) containing DNase and lysozyme, and lysed by sonication using 10 s pulses at 50 W for a total time of 10 min. Lysates were clarified by centrifugation at $50,000 \times g$ for 1 h at 4 °C, and the supernatant was used for affinity purification on HisTrap FF columns (GE Healthcare) according to the manufacturer's instructions. The 6His-Smt3 tag was removed by incubation with SENP1[415–644], and the 6His-GST tag was removed by incubation with 3 C protease. The proteins were simultaneously dialyzed in binding buffer. The liberated affinity tag and His-tagged SENP1 and 3 C proteases were removed by a second round of affinity purification with HisTrap FF columns (GE Healthcare). All proteins were purified by final size exclusion chromatography (HiLoad 16/600 Superdex 75 or 200 pg) in 20 mM TRIS pH 7.5, 150 mM NaCl, 2 mM dithiothreitol (DTT), concentrated using VIVASPIN 20 Columns (Sartorius), flash frozen in liquid nitrogen, and stored at −80 °C. Protein concentrations were determined by measuring absorption at 280 nm ($A_{280}$) using the extinction coefficients of the proteins derived from their sequences.

### Synthesis of activity-based probes

All activity-based probes used in this study were expressed as C-terminal intein fusion proteins as described previously[73]. In brief, the intein fusion proteins were affinity purified in buffer A (20 mM HEPES, 50 mM sodium acetate pH 6.5, 75 mM NaCl) from clarified lysates using Chitin Resin (New England Biolabs), following the manufacturer's protocol. On-bead cleavage was performed by incubation with cleavage buffer (buffer A containing 100 mM MesNa (sodium 2-mercaptoethanesulfonate)) for 24 h at room temperature (RT). The resin was washed extensively with buffer A, and the pooled fractions were concentrated and subjected to size exclusion chromatography (HiLoad 16/600 Superdex 75 pg) with buffer A. To synthesize the propargylated probe, 300 μM Ub/Ubl-MesNa was reacted with 600 μM propargylamine hydrochloride (Sigma Aldrich) in buffer A containing 150 mM NaOH for 3 h at RT. Unreacted propargylamine was removed by size exclusion chromatography and the probe was concentrated using VIVASPIN 20 Columns (3 kDa cutoff, Sartorius), flash frozen, and stored at −80 °C. NEDD8-PA was a kind gift from David Pérez Berrocal and Monique Mulder (Department of Cell and Chemical Biology, Leiden University).

### Chain generation

Wild-type Met1-linked di-ubiquitin was expressed as a linear fusion protein and purified by ion exchange chromatography and size exclusion chromatography. 6His-Met1-linked di-ubiquitin mutants R72A, R74A, M77A, Q78A, E92A, and E94A (M77A/Q78A/E92A/E94A correspond to M1(S1')/Q2(S1')/E16A(S1')/E18A(S1')) were expressed as linear fusion proteins and purified by affinity chromatography (HisTrap FF, Cytivia) and size exclusion chromatography. K11-, K48-, and K63-linked ubiquitin chains were enzymatically assembled using UBE2SΔC (K11), CDC34 (K48), and Ubc13/UBE2V1 (K63) as previously described[74,75]. In brief, ubiquitin chains were generated by incubation of 1 μM E1, 25 μM of the respective E2, and 2 mM ubiquitin in reaction buffer (10 mM ATP, 40 mM TRIS (pH 7.5), 10 mM $MgCl_2$, 1 mM DTT) for 18 h at RT. The respective reactions were stopped by 20-fold dilution in 50 mM sodium acetate (pH 4.5) and chains of different lengths were separated by cation exchange using a Resource S column (GE Healthcare). Elution of different chain lengths was achieved with a gradient from 0 to 600 mM NaCl.

### Ubiquitin-Nedd8/Nedd8-Ubiquitin chimeras

6His-Ubiquitin-Nedd8/6His-Nedd8-Ubiquitin chimeras were expressed as linear fusion proteins and purified by affinity chromatography (HisTrap FF, Cytivia) and size exclusion chromatography.

### Crystallization

Catalytic inactive SnOTU1–255 and linear linked di-ubiquitin were mixed in a 1:1.1 ratio and crystallized using sitting drop vapor diffusion with commercially available sparse matrix screens. 96 well crystallization plates containing 30 μl of the respective screening conditions were mixed with 10 mg/ml protein in the ratios 1:2, 1:1 and 2:1 in 300 nl drops. Initial crystals appeared after one week in JCSG A5 (0.2 M Magnesium formate, 20% w/v PEG3350) at 4 °C. These crystals were optimized by gradually changing the magnesium formate and PEG3350 concentrations using 48-well MRC plates with 80 μl reservoir solutions and 3 μl drops (protein/precipitant ratios: 2:1, 1:1, and 1:2). Further optimization was performed by addition of 0.3 μl of different additives to 3 μl protein/precipitant (Hampton Additive Screens I–III). Best diffracting crystals were harvested from a condition containing 0.25 M magnesium formate, 20% w/v PEG 3350 and 0.02 M Ethylenediaminetetraacetic acid disodium salt dehydrate and were cryoprotected with reservoir solution containing 20% glycerol.

### Data collection, phasing, model building, and refinement

The synchrotron data were collected at beamline P13[76] operated by EMBL Hamburg at the PETRA III storage ring (DESY, Hamburg, Germany). The data were processed using XDS[77]. An Alphafold[70] prediction was used as a molecular replacement search model, together with the ubiquitin coordinates from PDB entry 1UBI[78]. Molecular replacement was carried out using PHASER[79], as implemented in the phenix package[80]. First, two SnOTUs and two Ubiquitins were searched and found rapidly. We realized additional electron density and searched successfully for two more ubiquitin units. The initial models were refined using iterative cycles of phenix.refine[80] and manually rebuilt using COOT[81]. For structural analysis, PyMOL (http://www.pymol.org) and ChimeraX Graphics Systems[82] were used.

### AMC assays

Activity assays of DUBs against AMC-labeled substrates were performed using reaction buffer (150 mM NaCl, 20 mM TRIS pH 7.5, 10 mM DTT), 1 μM (or less if indicated) DUBs, 5 μM Ub-AMC (UbiQ-Bio, The Netherlands), or 5 μM SUMO1/SUMO2-AMC (Enzo Life Science). The reaction was performed in black 96-well plates (Corning) at 30 °C, and fluorescence was measured using an Infinite F200 Pro plate reader (Tecan) equipped with an excitation wavelength of 360 nm and an emission wavelength of 465 nm. The presented results are the means of three independent assays.

### Activity-based probe assays

DUBs were prediluted to 2× concentration (10 μM) in reaction buffer (20 mM TRIS pH 7.5, 150 mM NaCl, and 10 mM DTT) and combined 1:1 with 100 μM Ub-PA, SUMO1-PA, SUMO3-PA, ISG15[CTD]-PA, or NEDD8-PA for 18 h at 4 °C. The reaction was stopped by the addition of 2× Laemmli buffer and analyzed by SDS-PAGE using Coomassie staining.

### Ubiquitin chain cleavage/Ubiquitin-Nedd8/Nedd8-Ubiquitin chimera cleavage

DUBs were prediluted in 150 mM NaCl, 20 mM TRIS pH 7.5, and 10 mM DTT. The cleavage was performed at RT for the indicated time points with different DUB concentrations (as indicated in the respective figure legends) and 25 μM di-ubiquitin (M1, K11, K48, K63 synthesized as described above, K6, K29, K33 purchased from Biomol, K27 from UbiQ) or 25 μM Ubiquitin-Nedd8/Nedd8-Ubiquitin chimera. The reactions were stopped with 2× Laemmli buffer, resolved by SDS-PAGE, and stained with Coomassie stain.

## SUMO chain cleavage

SnCE1 was prediluted in 150 mM NaCl, 20 mM TRIS pH 7.5, and 10 mM DTT. The cleavage was performed at RT for the indicated time points with 5 μM SnCE1 and 5 μL Poly-SUMO-2 (SUMO2$_{2-7}$, 0.73 mg/mL, purchased from Enzo). The reactions were stopped with 2× Laemmli buffer, resolved by SDS-PAGE, and stained with Coomassie stain.

## Statistics and reproducibility

All activity-based probe, Ubiquitin/SUMO chain cleavage assays were performed at least two independent times with similar results. AMC assays were performed in triplicates. qPCRs were performed two independent times.

## Reporting summary

Further information on research design is available in the Nature Portfolio Reporting Summary linked to this article.

## Data availability

The X-ray structure of SnOTU/linear di-ubiquitin complex has been deposited at the PDB database under the accession number 8CMR. The X-ray structures of ubiquitin and Otulin are publicly available at the PDB database under the accession numbers 1UB and 5OE7, respectively. Source data underlying the findings of this study are provided with this article and its Supplementary Information. Source data are provided with this paper.

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

## Acknowledgements

We thank Christiane Horst, Claudia Poschner, and Kristýna Pekárková for expert technical assistance. The synchrotron data were collected at beamline P13 operated by EMBL Hamburg at the PETRA III storage ring (DESY, Hamburg, Germany); the excellent support of Dr. Michael Agthe is appreciated. Deubiquitinase research in the laboratory of KH is supported by DFG grant HO 3783/3-1. This work was also supported by a DFG GRK2243 grant to V.K.-P. Crystals were grown using equipment of the Cologne Crystallization facility (C$_2$f), which is supported by DFG grant INST 216/949-1 FUGG. We also thank the Regional Computing Center of the University of Cologne (RRZK) for providing support and computing time on the DFG-funded (Funding number: INST 216/512/1FUGG) high-performance computing (HPC) system CHEOPS.

## Author contributions

V.B. performed most biochemical experiments, T.H. contributed to biochemical experiments and performed the crystallizations, I.E. contributed experiments on SnVTD, M.U. collected the X-ray data and solved the structure, E.M.H. and V.K.P. performed bacteriological experiments, U.B. supervised the crystallography, K.H. initiated and supervised the study, and contributed bioinformatic analyses. All authors contributed to the data analysis and writing of the manuscript.

## Funding

## Competing interests

There are no competing interests in this manuscript.
