## [Peer Review File · Nature Communications]

REVIEWER COMMENTS

Reviewer #1 (Remarks to the Author):

In this manuscript Boll et al., characterize the different types of deubiquitinases present in the opportunistic bacterial pathogen *Simkania negevensis*. The manuscript characterizes the ubiquitin-chain Ubl specificity for all on them, and also in one example, they provide the crystal structure of the complex with a linear di-ubiquitin substrate. *Simkania negevensis* is a respiratory pathogen unrelated but with a similar lifestyle as *Legionella pneumophila*. Intriguingly both pathogens have developed high number DUBs to allow infection of the eukaryotic hosts. The authors describe the molecular mechanism by which SnOTU deubiquitinating enzyme possesses strong activity against linear Met1-linked ubiquitin chains, with a prominent role of the proximal ubiquitin interface. It is quite intriguing to know the role in this host-cell infecting bacteria of such a high number of DUBs to cleave all these different types of ubiquitin linkages, such as linear M1 or K6 chains. The results presented in this manuscript are quite interesting, the functional analysis of all SnDUBs and the structure of SnOTU-diUb are well done. The results will be of interest for the UbL-ubiquitin scientific community, in particular in the characterization of undescribed DUBs in bacteria.

A minor issues should be addressed.

In Supplementary Figure 1e. It's CDU1 or CDUB1 ?? I guess its CDU1.

In Supplementary Figure 1d. In the structural alignment of SnOTU, I would like to see the sequence of the N-terminal b-hairpin region, and check if there is homology with OTUB1 and OTUL. Also, it would be better to add a cartoon of the secondary structure above the alignment. Also the residue numbers.

In page 7, line 200. Please, explain better why it is not necessary to polarize the active site histidine in snOTU (lack of the acidic third residue). It is quite unusual to have only His and Cys in the active site (i.e. OTUB1 and OTUL contain a Asp or Asn). Perhaps there is another residue in the close proximity doing that role.

Because of the swapped dimer assembly in the asymmetric unit, several questions arise from the structure. The M1-diubiquitin cleavage site region is not clear to me, and there is no figure showing that. Do you see the location of ubiquitin Met1 ?. It is essential to have a Met1 in the reaction of snOTU ?. Do you have a M1A diUb mutant ?. Specificity for the linear M1-diubiquitin in SnOTU comes basically from the proximal ubiquitin binding ??

Also, could it be that in the crystal structure, an important interface around the catalytic region is missing ?? Namely the residues around the cleavage site (not observed because of this strange assembly).

Line 217, page 8. In supplementary figure 7, please show a general view of panel c. It is difficult to know where F4 and F25 are located.

Line 233, page 8. Ubiquitin Arg72 normally has an important role in DUBs, but here it seems the Arg74 contacts Glu183, and Arg72 is free. But the in vitro assays show opposite results. Why ?.

Reviewer #2 (Remarks to the Author):

In this manuscript, Boll et al. introduced several deubiquitinases from Simkania and examined their enzymatic characteristics by performing biochemical assays, including a ubiquitin chain cleavage assay to show linkage-specificity, and ub- or ubl-activity-based probe reactivity assays to show the specific activity as DUBs. They also solved a crystal structure of a novel-M1-specific DUB, SnOTU, in complex with M1-Ub2. Based on the crystal structure, the authors discovered several key residues, and the mutations on these residues identified key amino acids that give SnOTU M1-linkage specificity.

Taken together, this is an exciting story and gives new insights into DUBs from pathogenic bacteria. In addition, the new class of M1-specific DUB structure is also important for the people working in this field. However, while the authors carefully performed the experiments and exquisitely described their findings, there are a few concerns that should be addressed by the authors to further improve their outstanding findings.

Specific Comments:

- 1. (Sup. Fig. 2) The authors performed qPCR to examine the expression of SnDUBs during infection. While the authors concluded that some of them showed increased expression levels at the later stage of infection, it is still not clear whether these DUBs are expressed during infection. In addition, SnJos2 seems not to be expressed based on this data, but this DUB showed activity against multiple Ub-chains (Fig. 2e). While this manuscript is oriented toward biochemistry and structure, it is important to clarify whether these proteins are effectors and play roles during infection. Since the deletion of each SnDUB from Simkania and performing infection experiments require tremendous efforts and are a bit out of the scope of this story, I recommend that the authors carefully repeat the current experiment to check the mRNA level of each DUB and re-write the paragraph including the current knowledge about the Simkania or Chlamydia effector system.**
- 2. (Entire figures that have K63-linked Ub species) Throughout the entire figure, there are multiple bands appearing when the authors used K63-diUbiquitin. The authors need to clarify these bands.**
- 3. (In the Fig. 1b-d or other SnDUBs where authors used truncated versions) I suggest the authors put the exact boundaries of the proteins or mark them as SnUSP1core or SnUSP167-529 because the authors used full-length proteins for other SnDUBs.**
- 4. (In the Fig. 1c-d) It is an interesting point to mention in the manuscript why SnUSP2 and SnUSP3 interact with ISG15 while SnUSP1 does not. Can the authors distinguish any sequence-level conservation in these two DUBs? It would also be interesting to include a discussion point about ISG15.**
- 5. (Fig. 1e,f,g) The authors used entire Ub-linkages for SnUSP3 but not for SnUSP1 and SnUSP2. To conclude the linkage specificity, other chains need to be examined.**
- 6. (Sup. Fig. 3) Similar to the above point, it would be consistent to show the chain cleavage assay of SnUSP3 CtoA mutant, although this is not critical as the above point.**

7. (Line 108) Two SnJos1 and SnJos2 are assigned with the same gene code.
8. (Sup. Fig. 4) Proteins are labeled with ATX instead of JOS.
9. (Fig. 4c) Similar to point 2, why did the authors exclude K27 chain in the cleavage assay?
10. (Fig 5) The authors suggested that the SnOTU does not follow the canonical catalytic triad of OTU. I am just wondering whether it is an A20 type OTU that has a third acidic residue at the N-terminal to the catalytic cysteine. Based on the sequence in Sup fig 1, it can be E75 (the Glu of H'E'IQGDGNCL). (I am not sure about the numbering. The authors should add A.A. numbers in Sup Fig1.) How about the structure? Is there any possibility of E75 becoming a catalytic triad? Then, it would be interesting to introduce a mutation on it.
11. (Fig 5) With the mutations introduced on the S1' site, the authors explain that the S1' site on SnOTU1 makes more extensive contact with ubiquitin than the S1 site. Though the residual di-ubiquitin species are observed a bit more in S1 site mutations, it is still unclear which site determines the M1-specificity. I would suggest the author design an Ub-Nedd8 chimera and Nedd8-Ub chimera to examine the importance of ubiquitin recognition site. Well, this is still an optional experiment, and authors do not need to address this question if the authors think it is not relevant, but the authors need to tone down about the importance of S1' site with the current data as there are also critical amino acids such as E185 in S1 site as well.
12. (Fig 5) The authors solved the SnOTU1 in complex with M1-Ub2. I would recommend that the authors compare their structure with previously known M1-specific DUBs and show these figures in detail. This is relatively easy to perform but critical to emphasize how novel SnOTU is.
13. (Fig6c) The authors nicely performed the AMC-based cleavage assay, which is chemically different from the activity-based propargyl assay that the authors used for the entire paper. Now, I am just wondering why the authors did not perform this assay with other SnDUBs like SnOTU, SnJos. Though they did not react with propargyl, they might cleave AMC from ubiquitin.
14. (Fig 6d) Again, four chains are missing in the linkage specificity assay.

Reviewer #3 (Remarks to the Author):

Boll et al in this well-written manuscript present further evidence of the ever-expanding repertoire of bacterial deubiquitinases (DUBs) in intracellular pathogens. Recent studies in the last decade or so have brought to light the remarkable diversity of prokaryotic DUBs. The presence of evolutionarily diverse DUBs in prokaryotes is particularly striking given that they lack their own ubiquitin modification system. The DUBs are encoded by a variety of pathogens sharing an intracellular life style in eukaryotic hosts, which use the ubiquitin system in some form to detect and degrade the pathogen. Historically, the first reports described bacterial CE-clan DUBs, followed more recently by eukaryote-like OTU DUBs in several bacteria, and even a USP-family representative in Burkholderia. In this paper, the authors have used a specialized bioinformatical analysis to identify DUBs encoded in the

genome of an opportunistic bacterial pathogen, *Simkania negevensis*. Using an HMM-based search procedure they identified 12 putative cysteine-protease DUBs in this organism, most of them sharing a characteristic motif called the gatekeeper motif, a feature that distinguishes DUBs from other cysteine proteases. Using recombinant protein constructs they demonstrated DUB activity in eight of the predicted proteins through reactivity toward a cysteine-targeting Ub-electrophile probe and hydrolytic activity toward diubiquitin substrates representing different poly-ubiquitin chain types. The USP-family DUB identified generally showed robust activity in catalytic cysteine-dependent manner, albeit with no linkage preferences. They have found what could be the first Josephin-domain containing bacterial DUBs (SnJos1, SnJos2). Further, they identified a viral-tegument like deubiquitinase (VTD)-family DUB (SnDUB) showing a distinct preference for K6-linked diubiquitin and an OTU-family member (SnOTU) with a strong preference for M1-linked diubiquitin substrate. They were able to crystallize SnOTU with M1-diubiquitin substrate and capture essential feature of substrate recognition, providing a structural basis of its enzymatic activity.

Overall, this paper reports a number of significant results that may be of interest to biochemists, researchers interested in the ubiquitin system and the broad field of host-pathogen interaction. Through their work we learn that bacterial pathogens with an intracellular life style may make extensive use of proteins with deubiquitinase activity, perhaps to deal with host defense and other types of cellular remodeling. As they show here, the universe of prokaryotic DUBs can be quite expansive, with diverse evolutionary history. The discussion covering the parallels between *Legionella* and *Simkania* is insightful.

The biochemical evidence provided in support of DUB activity is largely satisfactory, albeit somewhat less so in cases involving Josephin DUBs, which show sluggish activity only under high enzyme concentration and prolonged reaction conditions, raising questions as to if their activity can be biologically meaningful. The paper does not touch on biological importance of any DUB activity in any significant way, which is fine, considering this report is mostly a foundational one introducing biochemical evidence in support of the existence of DUBs in what appears to be a somewhat obscure pathogen.

The authors may want to consider the following to improve the quality of their manuscript before publication.

1. Are the described DUBs also substrates of secreted apparatus?

This is a question worth answering in this paper. The authors provide some evidence of transcription of the genes in mammalian cells upon infection, but are these proteins injected through the secretion apparatus? I suspect the organism uses some sort secretion machinery to translocate effectors so that they could interact with the ubiquitin system of the host. Can the authors identify (by means bioinformatics) putative translocation signals? Most of the previously identified bacterial DUBs are secreted through either type-III or type-IV secretion system of their respective pathogen.

2. Gatekeeper aromatic residue

Since the authors relied on this motif as a guide in their bioinformatic search, one expects some experimental validation of the importance of this key element in DUB activity. The authors only describe the cysteine mutant (although in one example, they include the catalytic His as well), it would be worth asking if the gatekeeper aromatic residue is important for DUB activity. Along this line, it would be necessary to biochemically identify the other residues of the triad (or diad) across the entire set.

3. AlphaFold models

It would be interesting to see how the alphafold models of the DUBs look. Can these models shed some insights into the architecture of the catalytic domain? (for the USP ones, for example)

4. The CE-clan DUBs

I wonder if some of them also have acetyl-transferase activity? Along this line, it may be worth asking if the josephin DUBs have some other activity than deubiquitination (such as acetyl transfer), especially given that they lack the gatekeeper aromatic residue.

5. Strict K6 specificity

The DUB cleaves K27-diubiquitin too. It is a stretch to say that it is strictly K6-specific. I agree, one could say that the SnOTU is 'strictly M1-specific'.

6. I suspect some of the low activity cases could be due to oxidized catalytic cysteine. This could be checked in presence varying concentration of reducing agents. Or, high-resolution intact mass spec could provide some clues.

7. Crystal Structure with Diubiquitin

One would have expected to see molecular recognition involving Met1 of the proximal ubiquitin. Can the authors identify these residues on the enzyme?

8. In order to be consistent across the board, it would be good to include Ub-AMC assay for all the DUBs studies here (like Figure 6C).

Reviewer #1 (Remarks to the Author):

In this manuscript Boll et al., characterize the different types of deubiquitinases present in the opportunistic bacterial pathogen *Simkania negevensis*. The manuscript characterizes the ubiquitin-chain Ubl specificity for all of them, and also in one example, they provide the crystal structure of the complex with a linear di-ubiquitin substrate. *Simkania negevensis* is a respiratory pathogen unrelated but with a similar lifestyle as *Legionella pneumophila*. Intriguingly both pathogens have developed high number DUBs to allow infection of the eukaryotic hosts. The authors describe the molecular mechanism by which SnOTU deubiquitinating enzyme possesses strong activity against linear Met1-linked ubiquitin chains, with a prominent role of the proximal ubiquitin interface. It is quite intriguing to know the role in this host-cell infecting bacteria of such a high number of DUBs to cleave all these different types of ubiquitin linkages, such as linear M1 or K6 chains. The results presented in this manuscript are quite interesting, the functional analysis of all SnDUBs and the structure of SnOTU-diUb are well done. The results will be of interest for the UbL-ubiquitin scientific community, in particular in the characterization of undescribed DUBs in bacteria.

A minor issues should be addressed.

In Supplementary Figure 1e. It's CDU1 or CDUB1 ???. I guess its CDU1.

The correct name is CDU1. We have corrected Supplementary Figure 1e.

In Supplementary Figure 1d. In the structural alignment of SnOTU, I would like to see the sequence of the N-terminal β -hairpin region, and check if there is homology with OTUB1 and OTUL. Also, it would be better to add a cartoon of the secondary structure above the alignment. Also the residue numbers.

The panels in Supplementary Figure 1 generally only shows the conserved catalytic domains of the different DUB classes. In the case of SnOTU, the N-terminal β -hairpin is not part of the OTU domain and is not present in OTUB1 and OTULIN. To address the above request, we have generated an additional Supplementary figure 9, which includes the sequence of the β -hairpin region and the other secondary structure elements, together with some key residue numbers. At this position, the alignment also helps to appreciate the superposition of SnOTU and OTULIN requested by reviewer #2.

In page 7, line 200. Please, explain better why it is not necessary to polarize the active site histidine in snOTU (lack of the acidic third residue). It is quite unusual to have only His and Cys in the active site (i.e. OTUB1 and OTUL contain a Asp or Asn). Perhaps there is another residue in the close proximity doing that role.

We do not really know why SnOTU does not require an acidic third residue in its active site, but we clearly observe that our structure does not contain such a residue in a suitable position and yet is catalytically highly active. Generally, the acidic third residue appears less important in OTU-type DUBs than in some other cysteine proteases. As shown by Bailey-Elkin et al (Ref 40), some viral OTU DUBs also lack the third active site residue. Work from Jonathan Pruneda (Schubert et al 2020, Ref 25) and the Komander group has shown that several human and bacterial OTU DUBs tolerate mutations to their third active site residue without losing (much) activity. Here is a quote from Schubert et al:

“The acidic position is typically the second amino acid C-terminal to the general base His, and in similar manner to human OTUs, its mutation can result in complete, intermediate, or no loss of activity in the bacterial OTUs”.

We have expanded the text and added more references to make this point clearer (page 8 in the revised manuscript)

Because of the swapped dimer assembly in the asymmetric unit, several questions arise from the structure. The M1-diubiquitin cleavage site region is not clear to me, and there is no figure showing that. Do you see the location of ubiquitin Met1 ?. It is essential to have a Met1 in the reaction of snOTU ?. Do you have a M1A diUb mutant ?. Specificity for the linear M1-diubiquitin in SnOTU comes basically from the proximal ubiquitin binding ??

The general architecture of the structure, with Met-1 of the S1'-ubiquitin located next (but not connected) to the Carboxy-Terminus of the S1'-ubiquitin is now shown in Fig 5a and Supplementary Fig 7a. In the post-cleavage step captured in our structure, the Met-1 side chain does not make any critical sidechain contacts. As shown in the newly added Supplementary Fig 7e, di-ubiquitin mutated at Met1(S1') is cleaved as well as wildtype di-ubiquitin.

We have added more evidence to show that the linear specificity of SnOTU is indeed mainly due to proximal ubiquitin binding: As (also) requested by reviewer #2, we have created linear Ub-Nedd8 and Nedd8-Ub fusions and tested them for cleavage by SnOTU. While Ub-Nedd8 is not a substrate, Nedd8-Ub is cleaved efficiently (new supplementary Figure 7f,g). These results highlight that the S1'-recognition is the main determinant for SnOTU reactivity. We have added a paragraph describing these new results on page 9.

Also, could it be that in the crystal structure, an important interface around the catalytic region is missing ???. Namely the residues around the cleavage site (not observed because of this strange assembly).

To address this question, we have compared our structure to the AlphaFold model containing M1-linked di-ubiquitin. In addition, we have artificially introduced a peptide bond between the two ubiquitin units of our structure and analyzed the resulting structure changes after energy minimization and a short molecular dynamics time-course. In both analyses, no additional SnOTU-to-Ub contacts were observed, which have a chance of being catalytically important.

Line 217, page 8. In supplementary figure 7, please show a general view of panel c. It is difficult to know where F4 and F25 are located.

We have now changed panel 7c to show a general view.

Line 233, page 8. Ubiquitin Arg72 normally has an important role in DUBs, but here it seems the Arg74 contacts Glu183, and Arg72 is free. But the in vitro assays show opposite results. Why ?.

Question 1, why is Arg-72 important? In DUBs, Arg-72 of the S1 ubiquitin is often stabilized by a salt bridge to an acidic DUB residue. Our mutagenesis data (Figure 5k) show that Arg-72 is important for SnOTU reactivity, although in our structure there are no convincing salt bridge partners. The closest negative charge of Glu-185 is found at 4.4 Å distance. The importance of Arg-72 might be explained by an indirect effect via the π -stacking between Arg-72(S1) and Arg-42(S1), which in turn forms a salt bridge to Glu-185 of SnOTU. As shown in Fig 5j, Glu-185(OTU) is crucially important for activity, since the E185A mutant is inactive.

Question 2, why is Arg-74 less important? The refined structure shows a salt-bridge between Arg-74(S1) and Glu-183(OTU). However, according to electron density, both participating side chains are not well defined, casting doubt on the relevance of the salt-bridge. Moreover, not all observable salt bridges are functionally important. It is well possible that the bridge between Arg-74(S1) and Glu-183(OTU) is made redundant by the unique salt bridge between Arg-42(S1) and Glu-185(OTU) (Figure 5j).

All things considered, we cannot provide an unambiguous explanation for the relative importance of Arg-72 and Arg-74 and there is no easy experimental proof. Since we have extensively documented the superior importance of S1'-recognition over S1-recognition, we would rather not venture into excessive speculations on S1 recognition details.

Reviewer #2 (Remarks to the Author):

In this manuscript, Boll et al. introduced several deubiquitinases from *Simkania* and examined their enzymatic characteristics by performing biochemical assays, including a ubiquitin chain cleavage assay to show linkage-specificity, and ub- or ubl-activity-based probe reactivity assays to show the specific activity as DUBs. They also solved a crystal structure of a novel-M1-specific DUB, SnOTU, in complex with M1-Ub2. Based on the crystal structure, the authors discovered several key residues, and the mutations on these residues identified key amino acids that give SnOTU M1-linkage specificity.

Taken together, this is an exciting story and gives new insights into DUBs from pathogenic bacteria. In addition, the new class of M1-specific DUB structure is also important for the people working in this field. However, while the authors carefully performed the experiments and exquisitely described their findings, there are a few concerns that should be addressed by the authors to further improve their outstanding findings.

Specific Comments:

1. (Sup. Fig. 2) The authors performed qPCR to examine the expression of SnDUBs during infection. While the authors concluded that some of them showed increased expression levels at the later stage of infection, it is still not clear whether these DUBs are expressed during infection. In addition, SnJos2 seems not to be expressed based on this data, but this DUB showed activity against multiple Ub-chains (Fig. 2e). While this manuscript is oriented toward biochemistry and structure, it is important to clarify whether these proteins are effectors and play roles during infection. Since the deletion of each SnDUB from *Simkania* and performing infection experiments require tremendous efforts and are a bit out of the scope of this story, I recommend that the authors carefully repeat the current experiment to check the mRNA level of each DUB and re-write the paragraph including the current knowledge about the *Simkania* or *Chlamydia* effector system.

Indeed, it is currently not possible to genetically manipulate *Simkania negevensis*, precluding the possibility to experimentally prove the DUB candidates to be secreted effectors. To address the first part of the request, we have repeated the qPCR analyses for quantifying mRNA levels of the DUB candidates in human and amoebal infection systems. The data are shown in the expanded Supplementary Fig. 2 and are described on page 5 of the revised manuscript. As documented there, the relative expression levels of the various DUB candidates differ considerably between human and amoebal infections. To address the 2nd part of the request, we have added a new paragraph to the

discussion section (page 11) where we discuss Simkania secretion signals and the reasoning why the DUB candidates are likely effectors. In brief, Chlamydiales mainly use a T3SS for secreting effectors, which are difficult to predict from sequence alone. In the absence of recognizable secretion signals, the relationship of bacterial protein to typical eukaryotic sequence families is often considered a basis for effector prediction (Eichinger et al, Ref #51)

2. (Entire figures that have K63-linked Ub species) Throughout the entire figure, there are multiple bands appearing when the authors used K63-diUbiquitin. The authors need to clarify these bands.

This is a known property of our K63 preparations, which is also visible in our other DUB papers, e.g. Nat Commun 2022 Dec 10;13(1):7643, Nat Commun 2022 Jan 20;13(1):401, Nat Commun. 2018 Feb 23;9(1):799. We investigated the additional bands seen in uncleaved K63 di-ubiquitin (e.g. by intact-mass MS) and found them to represent reversible aggregates of di-ubiquitin, which completely disappear upon DUB cleavage. Thus, it is a purely cosmetic problem which is often addressed by simply cropping the gel. However, we prefer to show the entire gel image, which allows to assess and compare the DUB loading.

3. (In the Fig. 1b-d or other SnDUBs where authors used truncated versions) I suggest the authors put the exact boundaries of the proteins or mark them as SnUSP1core or SnUSP167-529 because the authors used full-length proteins for other SnDUBs.

We have added the exact boundaries to Figure 1b-d and all other similar figures.

4. (In the Fig. 1c-d) It is an interesting point to mention in the manuscript why SnUSP2 and SnUSP3 interact with ISG15 while SnUSP1 does not. Can the authors distinguish any sequence-level conservation in these two DUBs? It would also be interesting to include a discussion point about ISG15.

The sequences do not really indicate why SnUSP2 would react with ISG15 while SnUSP1 does not. Structural knowledge would be required so provide a good explanation. We have added a short paragraph to the discussion section (page 13) mentioning the ISG15-directed activity and speculating about its biological role.

However, we would rather not make a strong statement on the differences in ISG15 reactivity, since we did not investigate this aspect in sufficient detail. Our activity-based probes only encompass the 2nd half of ISG15; it is possible that full-length ISG15 would behave differently. Moreover, we know of several DUBs that are able to cleave ISG15 despite a complete absence of biological relevance. For example, the VTD-type DUB of Drosophila is very active against (human) ISG15 although insects do not encode ISG15-like modifiers.

5. (Fig. 1e,f,g) The authors used entire Ub-linkages for SnUSP3 but not for SnUSP1 and SnUSP2. To conclude the linkage specificity, other chains need to be examined.

Our original reasoning was that DUBs that already cleave K48, K63 and K11 chains are considered as linkage-promiscuous anyway, making it less interesting if they also cleave rare chain types. Nevertheless, we have now expanded the panels in Figure 1f-j to cover all chain types. We used lower DUB concentrations for the panels 1g and 1i to better visualize the subtle differences in

activity. The description of the new data has been added on page 5/6 of the results section.

6. (Sup. Fig. 3) Similar to the above point, it would be consistent to show the chain cleavage assay of SnUSP3 CtoA mutant, although this is not critical as the above point.

We have added these data as Supplementary Fig. 3f and mentioned it on page 6 of the results section.

7. (Line 108) Two SnJos1 and SnJos2 are assigned with the same gene code.

The Gene-IDs of SnJos1 and SnJos2 on page 4 (originally line 108) were double-checked. They are not identical, but very similar (SNE_A21920 and SNE_A21910) since the two genes are neighbors.

8. (Sup. Fig. 4) Proteins are labeled with ATX instead of JOS.

This was a remnant of our old nomenclature. We have updated Supplementary Fig 4.

9. (Fig. 4c) Similar to point 2, why did the authors exclude K27 chain in the cleavage assay?

We have added the missing data to Supplementary Fig. 6a and described it on page 7 of the results section. Please note that Supplementary figure 6c (the same assay performed with the SnOTU¹⁻²⁵⁵ construct used for crystallization) already contains a full panel of di-ubiquitin chains including K27.

10. (Fig 5) The authors suggested that the SnOTU does not follow the canonical catalytic triad of OTU. I am just wondering whether it is an A20 type OTU that has a third acidic residue at the N-terminal to the catalytic cysteine. Based on the sequence in Sup fig 1, it can be E75 (the Glu of H'E'IQGDGNCL). (I am not sure about the numbering. The authors should add A.A. numbers in Sup Fig1.) How about the structure? Is there any possibility of E75 becoming a catalytic triad? Then, it would be interesting to introduce a mutation on it.

In the available structure, the distance between a sidechain oxygen of Glu-75 and the closest ring-nitrogen of His-234 is 7.9 Å. The Glu-75 sidechain is held in place by a 3.4 Å salt bridge to Lys-45 and is thus unlikely to contribute to the active site.

As for the request of residue numbers in the snOTU part of Sup Fig1, we have included an expanded version of this alignment, including residue numbering and secondary structure elements, into the newly created Supplementary Fig 9a. At this place, the alignment also helps to appreciate the newly added superposition of SnOTU and OTULIN (see comment #12 below).

11. (Fig 5) With the mutations introduced on the S1' site, the authors explain that the S1' site on SnOTU1 makes more extensive contact with ubiquitin than the S1 site. Though the residual di-ubiquitin species are observed a bit more in S1 site mutations, it is still unclear which site determines the M1-specificity. I would suggest the author design an Ub-Nedd8 chimera and Nedd8-Ub chimera to examine the importance of ubiquitin recognition site. Well, this is still an optional experiment, and authors do not need to address this question if the authors think it is not relevant, but the authors need to tone down about the importance of S1' site with the current data as there are also critical amino acids such as E185 in S1 site as well.

We have addressed this issue by creating the linear mixed-chain constructs Ub-Nedd8 and Nedd8-Ub and testing them for cleavage by SnOTU. The results were very clear and support our claims: Ub-Nedd8 is a good substrate for S1-recognizing DUBs like USP21 but is not cleaved at all by SnOTU. By

contrast, Nedd8-Ub is not recognized by USP21 (due to the lack of S1-ubiquitin) but is a very good substrate for SnOTU (due to the presence of the S1'-ubiquitin). These results are documented in the new supplementary figures 7f,g and described on page 9. We feel that these data are a solid proof for the importance of the S1'-ubiquitin recognition.

12. (Fig 5) The authors solved the SnOTU1 in complex with M1-Ub2. I would recommend that the authors compare their structure with previously known M1-specific DUBs and show these figures in detail. This is relatively easy to perform but critical to emphasize how novel SnOTU is.

Currently, there are two M1-specific DUBs known, metazoan OTULIN and Legionella RavD. Of those, only OTULIN is an OTU-type enzyme and can be compared to SnOTU. We have added a new Supplementary Fig 9, which shows the superposition of SnOTU with human OTULIN and also a corresponding sequence alignment with residue numbers. A discussion of this figure and its implications has been added to the discussion section on page 13)

13. (Fig6c) The authors nicely performed the AMC-based cleavage assay, which is chemically different from the activity-based propargyl assay that the authors used for the entire paper. Now, I am just wondering why the authors did not perform this assay with other SnDUBs like SnOTU, SnJos. Though they did not react with propargyl, they might cleave AMC from ubiquitin.

We have added all the missing AMC assays and have added the results to Figures 1e, 2d, 3c, 4c. A description of the results has been added to the respective paragraphs.

14. (Fig 6d) Again, four chains are missing in the linkage specificity assay.

We have expanded Figure 6d to cover all chain types and describe the results on page 9 of the results section.

Reviewer #3 (Remarks to the Author):

Boll et al in this well-written manuscript present further evidence of the ever-expanding repertoire of bacterial deubiquitinases (DUBs) in intracellular pathogens. Recent studies in the last decade or so have brought to light the remarkable diversity of prokaryotic DUBs. The presence of evolutionarily diverse DUBs in prokaryotes is particularly striking given that they lack their own ubiquitin modification system. The DUBs are encoded by a variety of pathogens sharing an intracellular life style in eukaryotic hosts, which use the ubiquitin system in some form to detect and degrade the pathogen. Historically, the first reports described bacterial CE-clan DUBs, followed more recently by eukaryote-like OTU DUBs in several bacteria, and even a USP-family representative in Burkholderia. In this paper, the authors have used a specialized bioinformatical analysis to identify DUBs encoded in the genome of an opportunistic bacterial pathogen, *Simkania negevensis*. Using an HMM-based search procedure they identified 12 putative cysteine-protease DUBs in this organism, most of them sharing a characteristic motif called the gatekeeper motif, a feature that distinguishes DUBs from other cysteine proteases. Using recombinant protein constructs they demonstrated DUB activity in eight of the predicted proteins through reactivity toward a cysteine-targeting Ub-electrophile probe and hydrolytic activity toward diubiquitin substrates representing different poly-ubiquitin chain types. The USP-family DUB identified generally showed robust activity in catalytic cysteine-dependent manner, albeit with no linkage preferences. They have found what could be the first josephin-domain containing bacterial DUBs (SnJos1, SnJos2). Further, they identified a viral-tegment like deubiquitinase (VTD)-family DUB (SnDUB) showing a distinct preference for K6-linked diubiquitin

and an OTU-family member (SnOTU) with a strong preference for M1-linked diubiquitin substrate. They were able to crystallize SnOTU with M1-diubiquitin substrate and capture essential feature of substrate recognition, providing a structural basis of its enzymatic activity.

Overall, this paper reports a number of significant results that may be of interest to biochemists, researchers interested in the ubiquitin system and the broad field of host-pathogen interaction. Through their work we learn that bacterial pathogens with an intracellular life style may make extensive use of proteins with deubiquitinase activity, perhaps to deal with host defense and other types of cellular remodeling. As they show here, the universe of prokaryotic DUBs can be quite expansive, with diverse evolutionary history. The discussion covering the parallels between *Legionella* and *Simkania* is insightful.

The biochemical evidence provided in support of DUB activity is largely satisfactory, albeit somewhat less so in cases involving Josephin DUBs, which show sluggish activity only under high enzyme concentration and prolonged reaction conditions, raising questions as to if their activity can be biologically meaningful. The paper does not touch on biological importance of any DUB activity in any significant way, which is fine, considering this report is mostly a foundational one introducing biochemical evidence in support of the existence of DUBs in what appears to be a somewhat obscure pathogen.

The authors may want to consider the following to improve the quality of their manuscript before publication.

1. Are the described DUBs also substrates of secreted apparatus?

This is a question worth answering in this paper. The authors provide some evidence of transcription of the genes in mammalian cells upon infection, but are these proteins injected through the secretion apparatus? I suspect the organism uses some sort secretion machinery to translocate effectors so that they could interact with the ubiquitin system of the host. Can the authors identify (by means bioinformatics) putative translocation signals? Most of the previously identified bacterial DUBs are secreted through either type-III or type-IV secretion system of their respective pathogen.

This point is very valid and has also been addressed by reviewer #2 (see above). Since a genetic manipulation of *Simkania* is currently not possible, the problem cannot be addressed experimentally. The best thing we can currently do is provide circumstantial evidence.

We have added new mRNA expression data covering human and amoebal infection models (Supplementary Fig 2) and show that the expression of the DUB candidates is strongly influenced by the infection system (and/or time point). We have also devoted a new paragraph of the discussion section (page 11) to *Simkania* secretion systems and why we think that the DUB candidates are secreted effectors. In brief, *Simkania negevensis* encodes type III and IV secretion systems, whose signal sequences are not easily predictable (Eichinger et al, Ref #51). Some of the DUB candidates have recognizable T3SS signals, others have transmembrane helices. In the absence of recognizable secretion signals, the relationship of bacterial protein to typical eukaryotic sequence families is often considered a basis for effector prediction (Eichinger et al, Ref #51), a condition which is clearly fulfilled for the SnDUBs. Everything considered, we do not have proof that the newly discovered DUBs are secreted effectors, but this is the most likely function, since *Simkania* does not code for ubiquitin.

2. Gatekeeper aromatic residue

Since the authors relied on this motif as a guide in their bioinformatic search, one expects some experimental validation of the importance of this key element in DUB activity. The authors only describe the cysteine mutant (although in one example, they include the catalytic His as well), it would be worth asking if the gatekeeper aromatic residue is important for DUB activity. Along this line, it would be necessary to biochemically identify the other residues of the triad (or diad) across the entire set.

As requested, we have now added data i) for all catalytic histidine mutations and ii) for all mutations of the aromatic gatekeeper residues. The data were added to Supplementary Figures 3a-f, 4a-b, 6b, and 8a-b and are described in their respective results sections. VTD enzymes do not have gatekeeper motifs and thus no changes were introduced to Supplementary Fig 5. SnJos1 and SnJos2 also lack gatekeeper motifs although those are present in many other Josephins. To investigate if the loss of the critical aromatic residue is connected to poor activity of SnJosephins, we artificially introduced gatekeeper motifs, but did not find improved activity (Suppl. Fig 4a).

3. AlphaFold models

It would be interesting to see how the alphafold models of the DUBs look. Can these models shed some insights into the architecture of the catalytic domain? (for the USP ones, for example)

We have generated Alphafold models for all Simkania DUB candidates. For the USP-type DUBs, which have a substantial recognition surface for the S1 ubiquitin, Alphafold is able to model the catalytic domain with a single ubiquitin molecule properly inserted into the active site. The Alphafold models are provided as supplementary data files. The availability of these models is mentioned in the context of Supplementary Fig 1, which shows the alignments of the catalytic portions of Simkania DUBS.

4. The CE-clan DUBs

I wonder if some of them also have acetyl-transferase activity?

Several different activities have been identified for CE-clan enzymes, including DUBs, UBL-proteases, acetyltransferases and general proteases (Pruneda et al 2016, Ref #26). The focus of the current manuscript is on the DUB activity of the bioinformatically identified candidates, and we consider the functional characterization of the non-DUB members to be outside our scope. We know of preliminary data obtained by a collaborating group suggesting that the non-DUB enzymes SnCE2 to SnCE5 are **no** acetyltransferases. However, more confirmatory work is required. At this point, we would rather not cover this topic in any detail.

Along this line, it may be worth asking if the josephin DUBs have some other activity than deubiquitination (such as acetyl transfer), especially given that they lack the gatekeeper aromatic residue.

We agree that SnJos1 and SnJos2 are rather poorly active DUBs, but their activity is not worse than other established Josephins such as human ATX3 (Winborn et al 2008, Ref #39). For some examples, such as human JOSD1, an activation by posttranslational modification has been reported (Seki et al 2013, Ref #38). A similar mechanism might be possible for SnJos1/2 and would have eluded our analysis using bacterially expressed enzymes. We extended the discussion of this problem on page 14)

5. Strict K6 specificity

The DUB cleaves K27-diubiquitin too. It is a stretch to say that it is strictly K6-specific. I agree, one could say that the SnOTU is 'strictly M1-specific'.

We performed several experiments but never observed any K27 cleavage by SnVTD, although it might appear so from looking at Fig. 3d (formerly 3c). However, the mono-ubiquitin band observed in the K27 samples is already present before adding the DUB (compare time point 0) and does not increase during the 3h incubation period.

There is only one commercial source for K27 chains. Some of the batches used by us appear to contain more or less mono-ubiquitin. Note that we had to use different batches for the initial submission and revision of this manuscript. Thus, the amount of mono-ub found in Fig 3c differs from that in Supplementary Fig 6c.

6. I suspect some of the low activity cases could be due to oxidized catalytic cysteine. This could be checked in presence varying concentration of reducing agents. Or, high-resolution intact mass spec could provide some clues.

The inactive CE-clan enzymes SnCE2 through SnCE5 are more likely to have other substrates, as has been shown for many other CE-clan enzymes (Pruneda et al 2016, Ref #26). As mentioned above, the poor activity of the Simkania Josephins is also not too surprising when compared to ATX3 and other animal Josephins. Nevertheless, we addressed the possibility of activity loss by cysteine oxidation for the example of SnJos2. As shown in the figure below, increasing the DTT concentration (beyond the usual 10mM) or using TCEP or Mercaptoethanol did not affect enzymatic activity. We thus consider cysteine oxidation an unlikely explanation.

7. Crystal Structure with Diubiquitin

One would have expected to see molecular recognition involving Met1 of the proximal ubiquitin. Can the authors identify these residues on the enzyme?

We clearly observe a modular recognition of both ubiquitin units, with separate recognition residues for the proximal and distal ubiquitins. See figures 5f,g,j,k and the description in the results part. However, a specific recognition of the Met1 of the proximal ubiquitin is not involved (a similar question has been asked by reviewer #1, see above). In the complex structure, which is similar to a post-cleavage situation, the Met-1 sidechain does not make any critical contacts. As shown in the newly added Supplementary Fig 7e, di-ubiquitin carrying a M1A(S1') mutation is cleaved as rapidly as wildtype di-ubiquitin. The recognition of the proximal ubiquitin by the N-terminal β 1/ β 2-hairpin and the His-49/Asp-34 contact is crucial for binding the proximal ubiquitin; the distance of the S1'-binding site from the active site favors recognition of a linear di-ubiquitin over other chain types.

We have now added an important piece of evidence to emphasize the importance of proximal ubiquitin binding for the di-ubiquitin recognition: As shown in the new Supplementary Fig 7f-g, we have created linear Ub-Nedd8 and Nedd8-Ub fusions and tested them for cleavage by SnOTU. While Ub-Nedd8 is not a substrate, Nedd8-Ub is cleaved efficiently. It appears that only substrates with

ubiquitin at the S1' position are substrates for SnOTU. We have added a paragraph describing these new results on page 9.

8. In order to be consistent across the board, it would be good to include Ub-AMC assay for all the DUBs studies here (like Figure 6C).

A similar request has been made by reviewer #2, see above. We have added all the missing AMC assays and have added the results to Figures 1e, 2d, 3c, 4c. A description of the results has been added to the respective paragraphs.

Finally, the revised manuscript has been re-checked for grammar and language. A number of typos and other minor mistakes has been corrected. Those are not highlighted in the manuscript text to insure readability.

REVIEWERS' COMMENTS

Reviewer #2 (Remarks to the Author):

The authors have now addressed all of the questions that I have pointed out and responded to my concerns. They added new experimental data, statistically quantified data, and made corrections (in text, figures and tables) where necessary. I fully recommend the publication of this nice work.

Reviewer #3 (Remarks to the Author):

The revised manuscript is much improved thanks to the authors' sincere efforts in carefully addressing all the concerns raised by this reviewer and others. Insights presented in the manuscript will be useful in broadening our understanding of host-pathogen interaction involving ubiquitin modification.